# Improved estimation of diurnal variations in near-global PBLH through a hybrid WCT and transfer learning approach

Yarong Li[1,2,#], Zeyang Liu[3,#], Jianjun He[2,*]

[1]College of Earth and Environmental Sciences, Lanzhou University, Lanzhou, China

[2]State Key Laboratory of Severe Weaher Meteorological Science and Technology, Chinese Academy of Meteorological Sciences, Beijing, China

[3]College of Electronic and Information Engineering, West Anhui University, Lu'an, China

[#]These authors contributed equally to this work

[*]Corresponding author: Jianjun He (Email: hejianjun@cma.gov.cn)

## Abstract

Diurnal variations in planetary boundary layer height (PBLH) is highly linked to weather, climate, and environmental processes. However, remaining challenges persist in estimating its diurnal behavior at a large scale due to insufficient observations and limitations of operational retrieval algorithms. This study proposed a deep learning framework based on an attention-augmented residual neural network to estimate diurnal variations in near-global

PBLH, incorporating profiles from an non-sun-synchronous lidar (Cloud-Aerosol Transport System: CATS) and meteorological fields. The framework can largely address the issue of multi-layer structures in space-borne lidar signals, significantly improving the accuracy of PBLH retrieval during morning and evening (with accuracy improvement approach 30% compared to traditional algorithm). Due to insufficient observations aligned with CATS orbits,

a pre-train model was firstly trained using pseudo-labels from reanalysis, and then was transferred to observation-based target labels. The transfer model demonstrates superior performance in most regions and periods, outperforming classical algorithm in capturing PBLH magnitude and its diurnal variations. Further assessments over different land covers show that the transfer model estimated PBLH and diurnal patterns were highly consistent

with those from radiosondes, surpassing reanalysis outputs. For model capability, wavelet covariance transformation derived potential PBLH and temperature profiles emerged as dominant factors, with contributions exhibiting diurnal patterns. Overall, this work proposes a novel framework for large-scale PBLH estimation and provides insights for improving retrieval algorithms, particularly through integrating remote sensing and machine learning.

## 1. Introduction

The planetary boundary layer height (PBLH) plays key roles in land-air exchanges and lower tropospheric processes (LeMone et al., 2019; Medeiros et al., 2005), such as the transfer and exchange of heat, momentum, humidity, and materials (Garratt, 1994; Holtslag et al., 2013; Stull, 1988). As an interface between the turbulent boundary layer and the free atmosphere, PBLH acts as a significant barrier and represents the degree of turbulent diffusion, determining the upper limit of boundary layer processes and playing vital roles in weather, climate, and environmental studies (Che et al., 2019; Davy and Esau, 2016; Guo et al., 2021; Li et al., 2017). Particularly, weather and pollution conditions are largely dependent on the diurnal behaviors of PBLH, which dominates the atmospheric dispersion and vertical mixing of pollutants (Ding et al., 2013; Huang et al., 2023; Li et al., 2025).

Despite the crucial importance, accurately measuring diurnal variations of PBLH across large scaled areas remains challenging due to spatio-temporal limitations of current detection instruments. Radiosonde and lidar measurements allow precise representation of vertical atmospheric structure (Seidel et al., 2010; Seidel et al., 2012). The radiosonde derived PBLH generally serves as a benchmark for validating simulations, reanalysis, and remote sensing (Guo et al., 2021; Li et al., 2023; Yue et al., 2021).However, global radiosondes are generally launched two or four times per day, and its coverage is much sparse in less-developed regions (like Africa and South America). Lidar systems serve as a promising tool for continuous PBLH monitoring (Chen et al., 2022; Liu et al., 2021), benefiting from their operation at sub-minute temporal resolution. While ground-based lidar has limited spatial representation, space-borne lidar enables large-scale PBLH detection across diverse regions (Jordan et al., 2010; McGrath-Spangler and Denning, 2012). Li et al. (2023) demonstrated diurnal

variations in large-scale PBLH from an non-sun-synchronous satellite. However, the PBLH retrieved by them exhibited large deviations in accuracy and diurnal patterns due to uncertainties of retrieval and signal noises such as multi-layer structures.

Traditional algorithms for retrieving PBLH from satellite signals are typically developed either to detect abrupt jumps in backscatter profiles (Kumar et al., 2018; Liu et al., 2015) or to identify the first exceeding of an empirical threshold (Palm et al., 2021). These algorithms suffered from significant accuracy challenges, due to at least three limitations: (i) the presence of elevated residual layers prevent downward staring lidar from detecting the true PBLH; (ii) cloud contamination or advected aerosols induce noises into lidar echos; and (iii) parameter selection of algorithm affect its sensitivity to diverse profile structures. The primary challenge for retrieving the diurnal variation of PBLH perhaps lies in minimizing the influences of residual layers or multi-layer structures during its morning and evening transition periods (Su et al., 2020; Li et al., 2023). Numerous efforts have been taken to enhance the algorithm performance in operating multi-layer profile structures of space-borne lidar, such as utilizing graphic clustering (Liu et al., 2018) or implementing additional physical constraints (Kim et al., 2021; Su et al., 2017). However, to date, current algorithms have not yet achieved optimal performance, primarily due to their inability to effectively resolve ambiguity signal structures through automated detection.

In recent years, machine learning has been increasingly integrated into PBLH estimation, achieving evidenced improvements. Several studies have employed deep neural network frameworks to estimate PBLH using near-surface and vertical atmospheric variables (Nguyen et al., 2021; Su and Zhang, 2024), constructing non-linear mapping from meteorologies to PBLH. Based on parameters acquired from surface observations, remoter sensors, reanalyses, and simulations, several random forest models were developed to predict PBLH (Guo et al., 2024; Krishnamurthy et al., 2021), the results demonstrated greater consistency with radiosondes and effectively corrected some inherent biases. There are gradient boosting learning models been proposed (de Arruda Moreira et al., 2022; Peng et al., 2023), which sequentially fits multiple weak learners, allowing the model to learn iteratively and improve prediction accuracy progressively. These methodologies essentially address a regression

relationship between PBLH and associated meteorological variables. There are also machine learning models were employed to refine retrieval technique from only remote sensing data. Rieutord et al. (2021) compared an unsupervised (AdaBoost) and a supervised (K-means) learning, to judge whether the lidar signals originate from the boundary layer or the above free atmosphere. Mei et al. (2022) proposed a VGG16-based convolutional neural network for PBLH detection using wavelet covariance transformation (WCT) images of ground-lidar backscatter, which can effectively suppresses contamination from clouds and residual layers. Sleeman et al. (2020) improved PBLH measurement from backscatter profiles under cloud condition through convolutional network.

Existing machine learning methodologies exhibit significant advantages in capture PBLH and its diurnal variations from noisy lidar signals. However, these studies have almost been limited to ground-based sites, and either require additional meteorological variables affecting PBLH evolution to be provided or necessitate human intervention to process remote sensing signals. These site-scaled models may not be generalizable on larger regions or global scale. Few studies have focused on improving PBLH estimation from space-borne lidar through machine learning approaches. This is primarily due to training a robust model requires substantial feature samples been provided, yet ground-based observations aligned with space-borne lidar overpass orbits are extremely scarce, making it difficult to obtain sufficient target labels; while those unsupervised learning methods often fail to achieve the desired performance (Rieutord et al., 2021).

Given the aforementioned considerations, this study proposes to construct a temporally and spatially adaptive deep learning model to estimate PBLH and its diurnal variations on a near-global scale using space-borne Cloud-Aerosol Transport System (CATS). As the satellite operates on a non-sun-synchronous orbit, it can capture a complete diurnal cycle (Yorks et al., 2016). To address the issue of insufficient matching samples with satellite orbits, this paper employs a transfer learning strategy. The approach involves first establishing a pre-train model using large samples matched by reanalysis data. And then, the feature extraction capabilities of the pre-train model are transferred to small samples matched with ground truth values. By fine-tuning the model weights, its representation for real-world targets and

generalization will beenhanced, thereby constructing more accuracy measurements of diurnal variations in large-scaled PBLH. Overall, this work presents the first attempt to integrate attention mechanisms and transfer learning for diurnal PBLH estimation at near-global scale, overcoming the limitations of classical algorithms in handling multi-layer atmospheric structures.

## 2. **Dataset**

### *2.1 satellite-based lidar profiles*

This study aims to develop a robust and generalizable deep learning framework for PBLH estimation from space-borne CATS lidar. CATS is initiated to monitor atmospheric clouds and aerosols using advanced lidar technology and is mounted on the International Space Station's (ISS) Japanese Experiment Module. Launched on January, 10, 2015, the ISS operated in 51.6° inclined orbits at an altitude of ~405 km, covering tropical and mid-latitude regions. Unlike sun-synchronous satellite, CATS exhibits a repeat cycle of approximately three days and operates at non-fixed overpass times. These characteristics allow CATS to capture large-scale diurnal variations in aerosols (Yu et al., 2021) and clouds (Zhao et al., 2023), as well as in PBLH (Li et al., 2023) after approximately 16 days of running. Due to instrument malfunctions, available CATS backscatters for PBLH retrieval only limited from Mar. 2015 to Oct. 2017, exclusively at the 1064 nm. Such wavelength is more sensitive to aerosol structure and variations compared to 532 nm (Winker et al., 2007), but with a lower signal-to-noise ratio (SNR); such that CATS signals necessitate more rigorous de-noising processes. Herein, the study acquired 1064 nm 'Total_Attenuated_Backscatter' profiles (TAB) from the CATS V3.00 Level 1B product and 'Feature_Type' data from the Level 2 product. The collected L1B and L2 products have horizontal resolutions of 350 m and 5 km, while both maintain a vertical resolution of 60 m. Several additional CATS products: 'Profile_UTC_Time', 'DEM_Mean_Elevation', 'Bin_Altitude_Array', 'Opacity', 'Layer_Top_Bin', 'Layer_Base_Bin', 'Surface_Type', 'Sky_Condition' were collected to refine the input features when training model. Noting that only the daytime CATS products were

acquired, as the determination for nocturnal PBLH falls outside scope of this paper.

*2.2 Radiosondes and reanalyses derived PBLH*

    Given that radiosonde derived PBLH is typically recognized as ground truth, this study employed sounding profiles from Integrated Global Radiosonde Archive (IGRA) V2.0, which serves to generate benchmark PBLH and to assess performances of our deep learning model.

IGRA offers exceptional temporal and spatial coverage, with current 466 radiosondes sites (Fig. S1) available in CATS overpassing areas. We acquired IGRA data temporally aligned with the CATS orbits (2015–2017). Sounding profiles for PBLH determination encompass geo-potential height, temperature, dew point depression, wind speed and direction. The bulk Richardson number method (Vogelezang and Holtslag, 1996) was adopted here to calculate

the PBLH, which can even maintain enough effectiveness under stable atmosphere regime and coarse sounding resolutions. Nevertheless, procedures were still conducted to eliminate soundings with coarse vertical resolution: within 5 km from the surface, the profiles must include at least seven vertical levels of temperature and humidity records; along with at least four levels of wind records. If valid wind observations are fewer than seven levels, a cubic

spline interpolation was employed to fill missing values (Zhang et al., 2013). However, we should aware that radiosondes have standard launch schedule (fixed at two UTC), only a few soundings coincide with CATS orbits, spatio-temporal overlaps between the two CATS and radiosondes are quite scarce. Fig. S1 gives their match-up information, where relatively rough matching rules (with distance limited to 200 km and time to 1.5 hour) were performed

to enlarge the number of samples. As a result, we obtained totally 5368 valid matching samples, which cover the majority of the Earth's land, and larger sampling numbers observed in mid-latitude regions.

    While the robustness and reliability of radiosonde-based PBLH, using only 5,368 matched samples to train a model is far from sufficient, especially considering these samples fall

throughout diverse periods and areas. Therefore, two reanalyses outputted PBLH, ERA5 (the fifth generation European Centre for Medium-Range Weather Forecasts atmospheric reanalysis) and MERRA2 (the Modern-Era Retrospective Analysis for Research and Applications Version 2), were further acquired in this study. Two sets of PBLH have the same

temporal resolution (1-hour) but with discrepant spatial grids: 0.25° × 0.25° (ERA5) and
0.625° × 0.5° (MERRA2). The grid-based reanalyses were interpolated to the orbit-based
CATS data using inverse distance weighting to ensure they are spatially aligned. In this study,
the MERRA2 PBLH was employed to construct one of training sets for the model, partly
because it assimilates aerosol information compared to ERA5 (Gelaro et al., 2017), making it
more approach to the intrinsic nature of CATS retrieval. Our prior study also reported that
using classical algorithm retrieved PBLH from CATS was more consistent with that from
MERRA2 (Li et al., 2023). Moreover, we acquired vertical profiles of temperature, humidity,
and wind speed from MERRA2 website as meteorological features input to model. These
variables represent 3-hour averaged meteorological fields and were matched with CATS
orbits.

## 3. Methodologies

### 3.1 Generate training data

WCT is one of typical PBLH retrieval techniques from satellite-based backscatters. This
study employs the Haar wavelet transform (Gamage and Hagelberg, 1993):

$$W_f(a,b) = \frac{1}{a}\int_{z_b}^{z_t} B(z)h(\frac{z-b}{a})dz \tag{1}$$

where, $W_f(a,b)$ is the WCT coefficient, $a$ is dilation factor, $b$ denotes the central location of
vertical translation, $B(z)$ is backscatters, $z_b$ and $z_t$ represent the bottom and top limits when
integrating the Haar function, respectively. The Haar wavelet function is:

$$h(\frac{z-b}{a}) = \begin{cases} 1, & b-a/2 \le z \le b \\ -1, & b \le z \le z+a/2 \\ 0, & elsewhere \end{cases} \tag{2}$$

inherently, the WCT is designed to check the similarity between the lidar profile and wavelet
stepping function, its maximum peak represents the sharpest signal gradient, and thereby is
considered as PBLH. However, selecting a proper dilation factor is crucial, diverse dilation
values exhibit significant impacts on step WCT signals. Particularly, a smaller dilation cause
WCT being sensitive to small-scaled fluctuations in backscatter profile and is susceptible to

noise interference, whereas a larger dilation may smooth out thin aerosol layers.

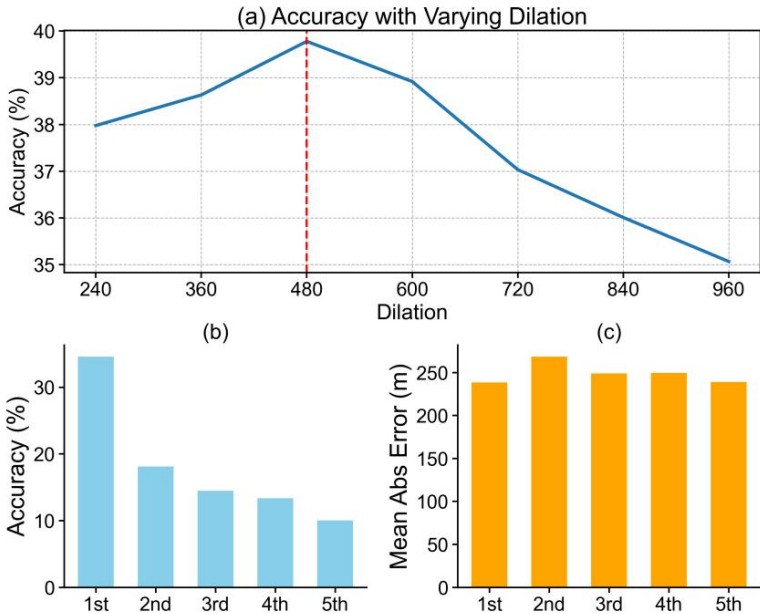

**Fig. 1**. Assessment for the WCT algorithm under different dilations (a); and accuracy (b) and MAE (c) compared against radiosonde derived PBLH when assuming one the first five peaks in WCT profiles (dilation=480) as PBLH.

Since varying sensitivity of different dilations to backscatter structures, we evaluated the retrieval accuracy of seven dilation values ranging 240–960 m (with an interval of 120 m) in Fig. 1a. Note that a tolerate PBLH bias of 500 m between WCT and radiosonde was utilized when calculating the retrieval accuracy, accounting for spatio-temporal matching errors and inherent algorithm differences between them. When compared against radiosonde derived

PBLH, a dilation of 480 m yielded the optimal results. Therefore, a dilation of 480 m is taken as a benchmark for WCT in this work. However, its maximum accuracy of 39.7% does not meet reasonable desire, such uncertainty is mainly induced by multi-layer structures such as residual layer and advected aerosols, and inability of WCT algorithm (Li et al., 2023). Here, the threshold of 500 m is only used to check whether the retrieval results of WCT are

close to the actual PBLH, rather than to illustrate the performance of the algorithm. Changing the threshold (Table S1) does not affect the key findings.

The WCT can, to some extent, be considered as a gradient-based algorithm, local peaks in WCT profile denote sharp changes in signal structure. A previous study adopted dynamic noise thresholds of ground-based lidar to identify the multiple layers (Li et al., 2023), but it is

not applicative to space-borne lidar profiles. Due to the magnitude of WCT represents the

intensity of local changes in backscatter profile, we hypothesize in this study that the local peaks in WCT profiles correspond to the top position of multi-layer aerosols; these peaks were then compared against the radiosonde derived PBLH (Fig. 1b). The results show that the first five peaks in WCT profiles align well with the truth PBLH, with their overall accuracy

exceeding 90% when we assumed one of these peaks to be PBLH. These peaks may not necessarily originate from the PBLH but may be induced by other interfering signals, whereas the first peak (i.e., benchmark for WCT algorithm), only capture few portion of truth PBLH. In other words, the WCT can effectively detect complex signal structures, while its maximum peak does not fully denote the PBLH. Therefore, the performance of WCT are

largely biased, particularly when it was utilized to CATS backscatters with strong temporal variability. Fig. 1c further examine mean absolute errors (MAE) when assuming one of the first five WCT peaks as PBLH, the MAE values (~240 m) are much lower than that using WCT algorithm (~1 km, not show here). Moreover, the hit rates and MAEs across multiple peaks under various dilation parameters also indicate that selecting 480 m as the dilation for

WCT is the most appropriate for this study.

Consequently, this study proposes to develop a deep learning framework to identify the optimal peak from the first several peaks of WCT profiles that aligned with the truth PBLH. Three types of feature data: remotely sensed profiles, meteorological profiles, and auxiliary parameters serve as model inputs. We used the raw CATS backscatter profile as one of the

remotely sensed features. Due to the lower SNR, a series of pre-processing procedures were implemented. First, we utilized the 'Opacity' parameter to remove opaque profiles, ensuring downward scanning CATS lidar can detect entire atmosphere columns. Samilar as previous retrieval practices (Li et al., 2023), profiles containing cloud layers were filtered using the 'Feature_Type' and corresponding 'Layer_Top_Bin', 'Layer_Base_Bin' from CATS L2 product.

Since CATS L1B and L2 products have diverse horizontal resolutions (a single L2 profile involves 14 L1B profiles), all of 14 L1B profiles would be eliminated if any cloud layer exists in the L2 profile. Noting that cloud screening was only applied below 5 km, profiles remained available when the lowest cloud base exceeded this altitude. Prior studies have suggested that long-distance horizontal smoothing can enhance SNR of daytime CATS

profiles (Nowottnick et al., 2022; Palm et al., 2021). Accordingly, the L1B profiles were then horizontally averaged across 60 km, meaning each training unit aggregated 60 km of raw CATS profiles. However, the solid ground generally return stronger signal echoes than the above aerosols, which could potentially distort the long-distance smoothing. To address this, we re-aligned the CATS profiles according to their elevations, ensuring consistent bin for

ground layers in a single training unit. Moreover, elevations of CATS profile extracted from the 'DEM_Mean_Elevation' may slightly bias from the true ground level, we thereby followed the same approach as Li et al. (2023) to re-calibrate the ground bin. Finally, to prevent the model from learning unforeseeable signal noises, we adopted a vertical smoothing window spanning three vertical bins into the profiles.

Based on the above cloud-screened, re-aligned, and horizontally averaged CATS profiles, we calculated the corresponding WCT profiles based on a dilation of 480 m. This study limits the PBLH estimation to height below 5 km (corresponding to 84 CATS bins), which covers the vast majority of global cases. Additionally, the two lowest bins (nearest the surface) were excluded to minimize ground noise contamination. Consequently, the derived PBLH values

range from 360 m ($120 + a/2$) to 4800 m ($5040 - a/2$). From each WCT profile, we acquired an additional profile involves the candidate PBLH, with the same shape as the backscatter and WCT profiles. In candidate profile, most bins were assigned as '0', while the bins corresponding to local WCT peaks were marked as '1'. The WCT peaks were selected based on their sorted magnitudes, with a maximum of five peaks retained per profile. Overall, three

remote sensing based profiles, encompassing averaged TAB, WCT, and candidate PBLH, each with dimensions of $84 \times 1$, are incorporated as model inputs.

The meteorological profiles include temperature, relative humidity, and wind speeds obtained from MERRA2 reanalysis,which were first matched with each CATS orbit, following inverse distance weight for spatial matchup and most proximity for temporal

matchup. And then the spatio-temporally matched MERRA2 profiles were vertically aligned to 84 CATS bins using a linear interpolation method.. In addition, the model inputs incorporated several non-profile parameters extracted from CATS auxiliary products, including geography information (latitude, longitude, altitude), local standard time (LST;

converted from UTC of each profile), surface type (based on MODIS land cover catagories) , and sky conditions. These non-profile parameters are one-to-on attached to CATS profiles and were subsequently resampled to match the dimensions of the profile features, and finally forming a standardized input array (84 bins × 12 features) for training the model, as the input layer shown in Fig. 2.

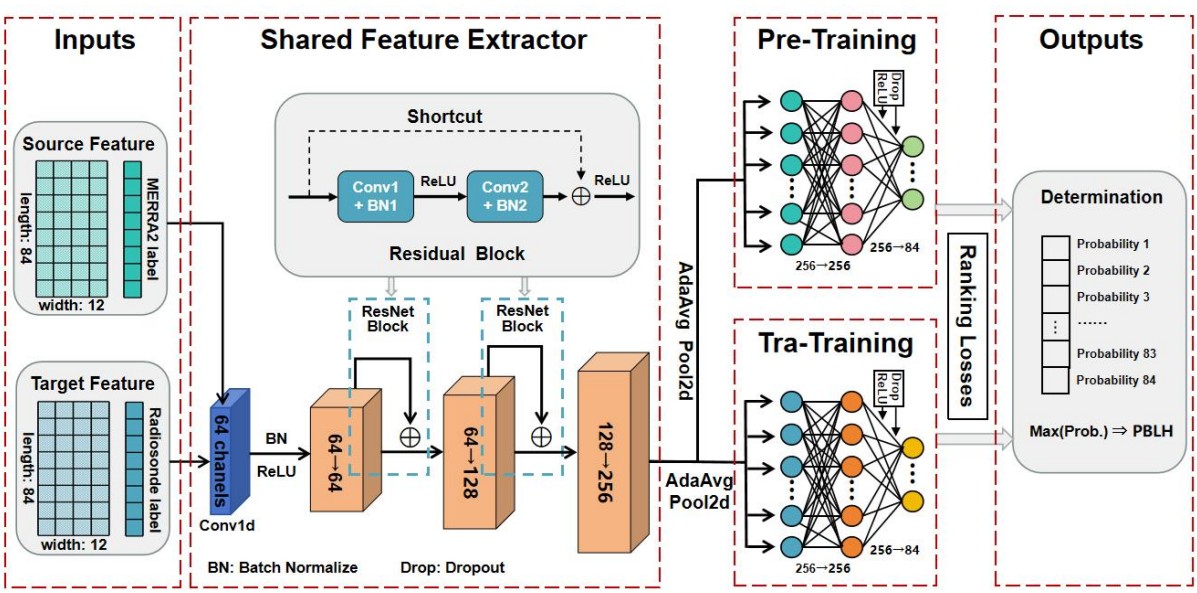

**Fig. 2**. Visualization for the model architecture, encompassing input layer (pre-training set and transfer training set), shared feature extractor (involving two attention augmented residual blocks), prediction heads (two discrepant fully connected layers for pre-training and transfer-training), and output layer.

In principle, the target labels for model training should be generated from radiosonde derived PBLH. However, this study obtained only 5,368 matched samples between CATS and radiosonde data, which are far too limited to train a model capable of capturing both temporal and spatial PBLH variations. To address this challenge, a transfer learning strategy was adopted. Specifically, a base model was pre-trained across a large feature set using pseudo-labels constrained by MERRA2 PBLH, after which the pre-train model was fine-tuned on a smaller dataset with target labels constrained by radiosonde derived PBLH. During the pre-training phase, the target lable was defined as the single peak in the WCT profile closest to the MERRA2 PBLH, allowing a maximum deviation of 480 m that equals to one-fold dilation value. This approach enables the model to learn vast feature information and substantially expands the training sample size. For pre-training, a feature dataset of 2016 covering a completed calender year was employed, comprising 113,488 samples in total, and

were split into training (80%) and validation (20%) subsets. In the transfer-learning stage, the target labels were constrained by radiosonde derived PBLH. There are 5008 feature samples were extracted from the matched CATS-radiosonde samples. Of these, 4258 samples were used for transfer training, while the remaining 750 samples served as a common testing set to assess model performances for both pre-training and transfer-learning stages. It is worth

noting that the 750 samples in test set were not randomly chosen. We carefully consider the sample size and distribution to ensure that they could cover most of the space and time, while also ensuring that there is no data leakage. In fact, for matching samples from different orbits, the possibility of data leakage is extremely low due to the time and space isolation. However, for the same orbit, if the distance between two samples is too close, there may be a data

leakage risk. Therefore, all samples on the same orbit that are within 300 km of other samples were placed in the training set while not in the testing set.

*3.2 Model architecture*

    A residual neural network (ResNet) attempts to learn the residual mapping between input

features and outputs, effectively alleviating the vanishing gradient problems in a deep neural network. This study constructed a ResNet based transfer learning framework for target location detection, aiming to identify the only bin representing the PBLH. The approach reshaped inputted feature array and employed a deep neural network to estimate the probability of each bin approximating the truth PBLH. As illustrated in Fig. 2, the model

adopted a modified ResNet-18 architecture (He et al., 2016), consisting of four main components. (i) Input layer: the model receives 2-D feature vectors ($84 \times 12$) without spatial reshape, maintaining the original temporal structure. (ii) Initial feature extracting: a 1-D convolutional layer with 64 channels (kernel_size=7) processes the input sequence, followed by batch normalization and ReLU activation. This maintains the original sequence length

while expands the channel dimension. (iii) Attention augmented residual blocks: three groups of down-sampling networks ($64 \rightarrow 128 \rightarrow 256$ channels) process the extracted features, containing two residual blocks. Notably, all convolutions use kernel_size=3 with to preserve sequence length. Each residual block incorporates a parallel attention mechanism, where the

positions of candidate PBLH are transformed through a 1-D convolution to weight the feature maps. Skip connections are implemented through $1 \times 1$ convolutions when channel dimensions change. (iv) Prediction heads: our model architecture includes a global average pooling across the temporal dimension to aggregate sequence information, and two fully connected layers ($256 \rightarrow 256 \rightarrow 84$) with ReLU activation and dropout. Sigmoid activation producing probability scores for each bin, the losses during training process were ranked to ensure that the score of target bin is higher than that of non-target bins.

The architecture involves an end-to-end supervised learning approach to train an enhanced attention-based ResNet based on PyTorch framework, where candidate PBLH with single channel was mapped to 64 channels via 1D convolution to align with the main ResNet networks, transforming position information into attention weights that explicitly leverage prior knowledge for improved PBLH prediction. For the hyper-parameter tuning, the model was trained using the Adam optimizer with an initial learning rate of 0.001, and was optimized via binary cross-entropy loss. To prevent over-fitting, a dropout regularization with a rate of 0.3 was implemented in the last fully connected layers, and an early stopping mechanism was enabled (patience=10). Training process would be terminated when the validating accuracy does not improved for 10 epochs.

Transfer learning is an efficient deep learning strategy that leverages prior knowledge from pre-train model to address new tasks (Pan and Yang, 2010). In this study, we first trained a ResNet model as our base network on larger samples with target labels constrained by MERRA2 PBLH. By virtue of the strong feature extraction capability of the pre-train model to learn common hierarchical features from the input data, we then transferred it to a new task, establishing the optimal prediction model. For this new task, the classification head at the end of the pre-train model was removed and replaced with new fully connected layers, the weights of the third residual block were also released, which were re-trained on the smaller transfer-training dataset. Meanwhile, the weights of other convolutional layers were kept frozen to preserve the learned feature representations. During transfer training, we employed a fine-tuning strategy with a lower learning rate (0.0001), reduced training epochs and early stopping tolerance to prevent overfitting.

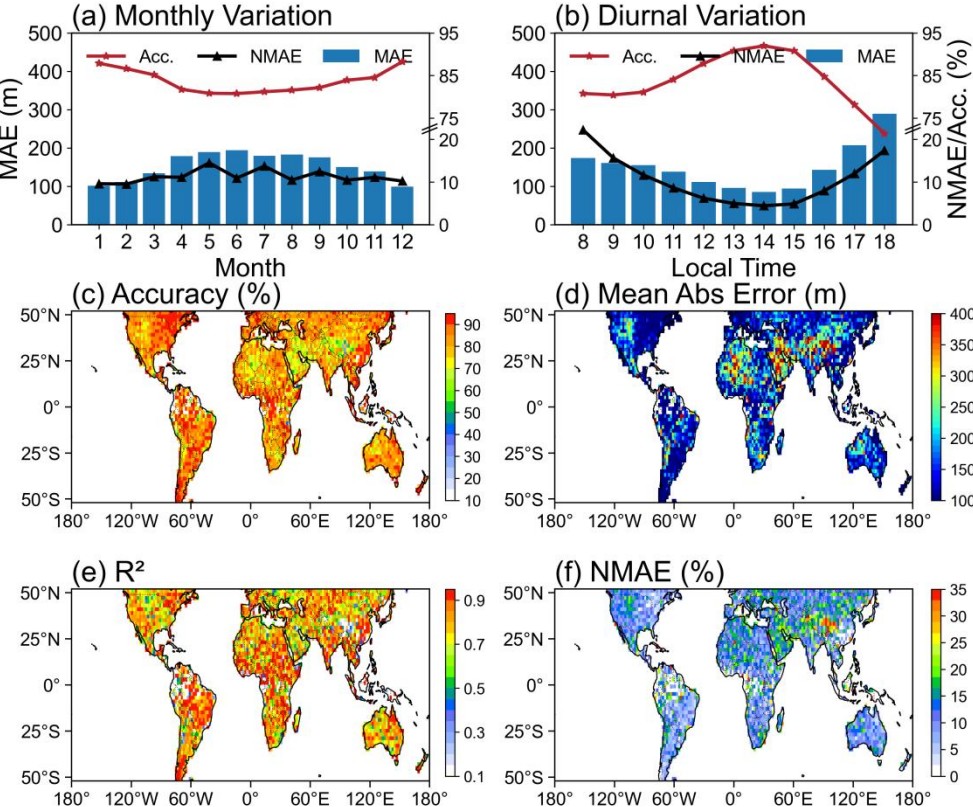

**Fig. 3**. Assessment of the pre-train model. (a-b) give the accuracy (column), MAE (black solid line) and NMAE (red solid line) at monthly and hourly scale, respectively; (c-f) denote the spatial distributions of accuracy, MAE, $R^2$, and NMAE, respectively.

Given the majority of matched CATS-radiosonde samples fall in land (Fig. S1), features over oceans were filtered out when training the pre-train model. As training curves shown in Fig. S2, the pre-train model achieved its optimal validation accuracy at 38th epoch, and training was stopped at 48th epoch due to early stopping. The optimal model demonstrates an accuracy of 80.24% on the training set and 81.18% on the validation set, with corresponding losses of 0.0209 and 0.0204. Over the common testing set, the pre-train model achieves an accuracy of 59.2%. Such testing accuracy is slightly lower when compared the base model (training model only in the radiosonde constrained training set, with testing accuracy of 60.0%), see training curves in Fig. S3. However, our transfer training achieves an better performances than both the pre-train and base models. The transfer model early stopped at 27th epoch (Fig. S4), reaching accuracies of 72.85% and 71.79% over training and validating sets, and a testing accuracy of 68.3%. This indicates that employing a transfer learning strategy can effectively enhance the model's learning capabilities and increase its

generalization.

Fig. 3 preliminarily evaluates the temporal (monthly and hourly) and spatial differences in accuracy, MAE, determination coefficient ($R^2$), and normalized mean absolute error (NMAE) of the pre-train model. The results indicate the pre-train model performed well over most land areas. However, the model's representation in high-altitude regions (Tibetan Plateau, Rocky Mountains) and desert areas (Sahara, Arabian Peninsula) are somewhat weak, where the accuracy drops below 80% and the MAE exceeds 400 m, particularly the $R^2$ and NMAE denote the model's performances are quite poor over complex terrains. These inabilities can be attributed to three main causes. First, the long-distance signal smoothing in processing raw CATS profiles may cause uncertainties over complex terrain. Second, grid-based MERRA2 data represents average state within a grid-cell, potentially leading to matching errors with orbital CATS observations in high-altitude areas. Finally, meteorological profiles and PBLH from MERRA2 may contain pronounced errors in these regions due to sparse observations available for assimilation.

The capabilities of pre-train model also exhibits seasonal and diurnal discrepancies. Particularly, the model demonstrate poorer performance from April to September compared to other months. As the poorer performances are primarily sourced from the Northern Hemisphere, it can be concluded that the model's representation in spring and summer seasons are somewhat weaker than that in autumn and winter. For the spring and summer seasons, the atmosphere is vigorous, accompanied by more convective activities. This leads to more complex aerosol structures (more noised CATS signal), but also limits the representation ability of MERRA2. In contrast, the atmosphere is more stagnant, and the aerosol structure is simpler (Li et al., 2025). Additionally, our assessment is mainly based on absolute deviations. The higher PBLH magnitude in the spring and summer seasons will cause the assessment being worse. When considering relative deviation (NMAE, Fig. 3a), the performance improves somewhat, but it is still slightly poorer than that in the autumn and winter. From a perspective of diurnal variation, the pre-train model performs less effectively during morning and later afternoon hours compared to around midday, with particularly poor performance observed in the later afternoon.

 *3.3 Feature importance permutation*

Based on the transfer model, we examined the importance score of each input feature using permutation importance technique (Altmann et al., 2010; Breiman, 2001). By randomly shuffling individual feature and measuring decreases in model performance, this method directly quantifies feature importance and can capture the non-linear dependencies among different features. Since the proposed ResNet model is essentially a classification task, we quantified the feature importance scores by calculating the increase in MAE induced by feature shuffling. Specifically, permutation importance estimations were implemented based on radiosonde constrained dataset (5008 samples), and the baseline MAE over original testing set was firstly derived. And then, we randomly shuffled the target feature across all samples, ensuring that 84 bins of target feature move synchronously from every input sample, while keeping other features unchanged. This will break the association between the target feature and predict label and is much applicable for our position sensitive predict task. The importance score is determined by the increased magnitude of MAE after permutation, a larger MAE increase indicates an higher feature importance. To enhance the robustness of feature permutation, each feature undergoes 30 independent iterations with different random sequences, noting that the input features were shuffled using a common random seed at each iteration. The ultimate importance scores were represented as mean value across 30 iterations.

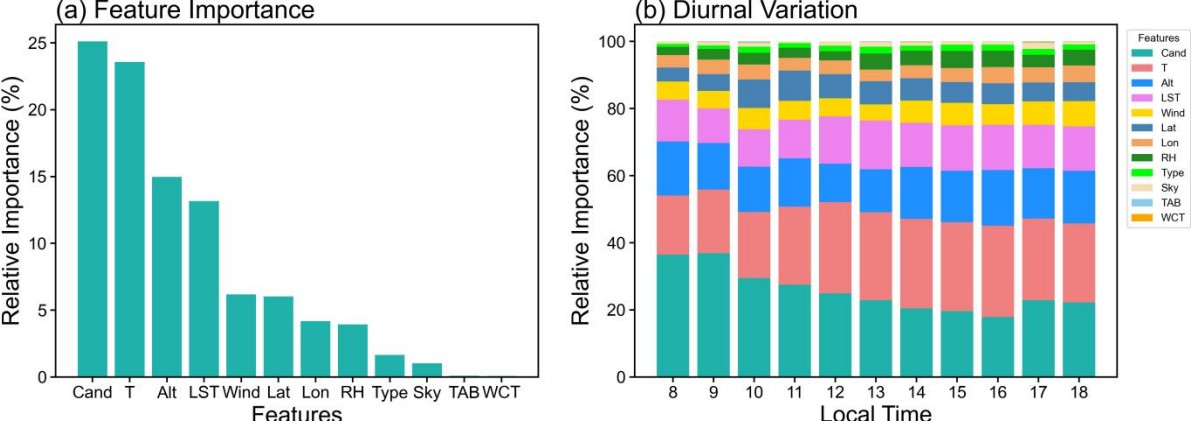

**Fig. 4**. The permutation importance of input features is measured by the increase in MAE when each individual feature is randomly shuffled. These importance scores are then normalized to represent their relative contributions (a), with the total importance summing to 100%. (b) illustrates their relative

importance scores at each hour.

The obtained importance score of each input feature was recalculated to derive its relative contribution rate. As shown in Fig. 4a, two profile features (candidate PBLH, temperature), along with two non-profile features (alititude, LST) emerge as the most important features, each with relative importance exceeding 10%. Geographic associated variables (latitude, longitude) and two meteorological (humidity, wind speed) profiles contribute the secondary importance, collectively contributing over 25% to the total importance, whereas surface type and sky conditions contribute marginally. Among the three remotely sensed profiles, importance scores of TAB and WCT are negligible, despite candidate PBLH playing the dominant role in the model. This implies that local peak/valley locations in backscatter profiles are more important than other shape features when estimating PBLH from CATS profiles. This may also suggest potential direction for improving classical retrieval algorithms of PBLH. That is, the shape and structure of remotely sensed profiles provide limited information about the PBLH, efforts should be taken to incorporate other diagnostic data, as also suggested by (Su et al., 2020). This also promote an implication for refining performances of classical algorithm, many of signal structures in the lidar profiles are noisy and meaningless. Instead of further refining profile-shape as our previous study (Li et al., 2023), incorporating thermodynamic and terrain-related diagnostics appears more beneficial.

We further extracted the permutation importance of input feature at each hour, and present their diurnal variations (Fig. 3b). The hourly importance scores of the two dominant contributors (candidate PBLH and temperature) vary evidently, while the diurnal variations of other importance scores are relatively slight. The combined importance of the two dominant contributors exceeds 45%, and their diurnal variations exhibit an alternating dominance pattern. Specifically, candidate PBLH dominates the model's capability during the morning periods with a gradually decreasing tendency, while the temperature emerges as the primary factor in the afternoon, with its importance scores essentially surpassing those of candidate PBLH. The diurnal variations in these importance scores might lead to diurnal behaviors of model performance (Fig. 3b), which will be discussed in the next section.

## 4. Results and Discussions

### 4.1 Assessing the model

Herein, we evaluated the performance of the transfer model by checking whether the model effectively captures the target labels constrained by radiosonde derived PBLH. Fig. 5 illustrate the spatial distributions of accuracy, MAE, and NMAE for transfer model, as well as their diurnal variations for WCT, base, pre- and transfer models. transfer. Notably, the calculated accuracy for WCT is slightly higher than that in Fig. 1a, because the current assessment is carried out only on the feature samples (5008) rather than all of the matched CATS-radiosonde samples (5368). All the three metrics denote the transfer model's prediction ability is weak in Western Asia and western North America, which are regions (Fig. 3) characterized by high-elevations and deserts. The pre-train model also performs poorly in these regions, partly because both the meteorological and lidar profiles over these regions have relatively low data quality (Li et al., 2023). Overall, the pre-train and transfer models demonstrate different degree of enhancements related to the classical WCT algorithm, and the performance of transfer model is reasonably better than pre-train and base model. As shown in Fig. S5, the transfer model achieves an improvement at nearly all sites. Quantitatively, the transfer model achieves an increase of 27.7% in accuracy and a reduction of 596.5 m (55%) m in MAE (NMAE) compared to the WCT,, demonstrating the substantial advantage of transfer training in refining driunal PBLH measurements from CATS data.

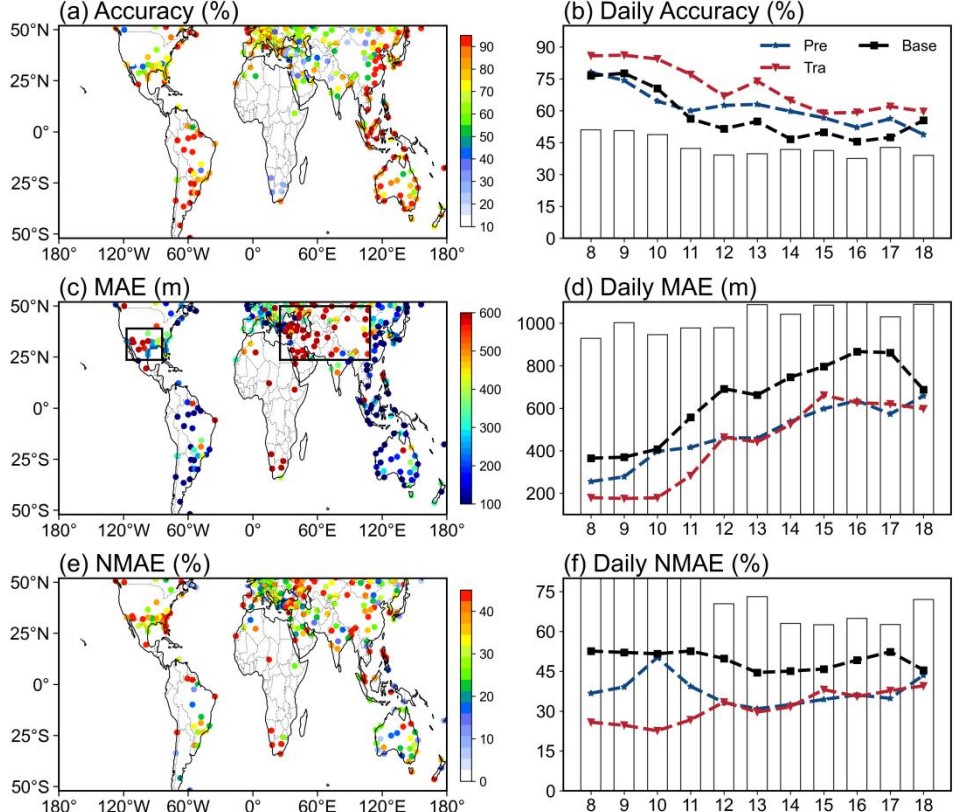

**Fig. 5**. Performance comparisons of the WCT, base, pre-train, and transfer model against radiosonde constrained target labels. (a, c, e) show the spatial distributions of accuracy, MAE, and NMAE for transfer model,    (b, d, f) display the diurnal variations of these metrics for WCT (column), base (back dash), pre-train (blue dash), and transfer (red dash) models. transfer

For the diurnal variations, transfer model performs well during the morning and midday periods but poor in the afternoon. In other words, its performance deteriorates over daily hours. It is interesting to note that the diurnal variations of the model performance align closely with the importance scores of candidate PBLH in Fig. 4b, while exhibits an inverse tendency with that of temperature. This further underscores the dominant roles of these two factors in regulating the model's capability. These diurnal variations may be largely regulated by the spatial distribution of training samples. Since radiosondes are only launched at two fixed times (00:00 and 12:00 UTC), these sites can provide samples at different local time. The lowest accuracy and largest MAE/NMAE typically occurred during 14:00–16:00 LST, with most samples originated from western North America and the Western Asia (see rectangular boxes in Fig. 5c).Since the PBLH magnitude over these regions is generally higher than others, an absolute error may bias the assessment; however, the relative error (i.e., NMAE) also demonstrate the model's ability is weak in afternoon.transferAdditionally, the

pre-train model exhibits generally weak performance during morning and later afternoon periods (Fig. 3b), whereas the transfer model performs better in the morning than at other daily times. This may attribute to the fact that morning samples are predominantly collected from areas around 120° E and 60° W, where the pre-train model performs stronger feature extraction capabilities in these low-altitude areas compared to others (Fig. 3c-d).

transfertransfer

*4.2 Inter-comparison of multi-sourced PBLH*

The above analyses primarily involve to validate the model's capability in capturing target labels, where the positions typically correspond to the WCT peak closest to either MERRA2 or radiosonde derived PBLH. In fact, the core function of the model is selecting, from

multiple WCT peaks, the one that most accurately represents the PBLH based on meteorological and physical conditions. It is crucial to aware that the model output remains a remotely sensed product, while radiosonde derived PBLH is regarded as closest to ground truth and generally serves as benchmarks for validating other measurements. Accordingly, Fig. 6 presents scatter plots comparing PBLH estimated by WCT, base model, pre-train

model, transfer model, MERRA2, and ERA5 against those from radiosondes. To enable systematic comparisons, their outputs within 200 km of sounding sites were averaged to derive statistical metrics, including correlation coefficient (R), MAE, NMAE, and root mean square error (RMSE).

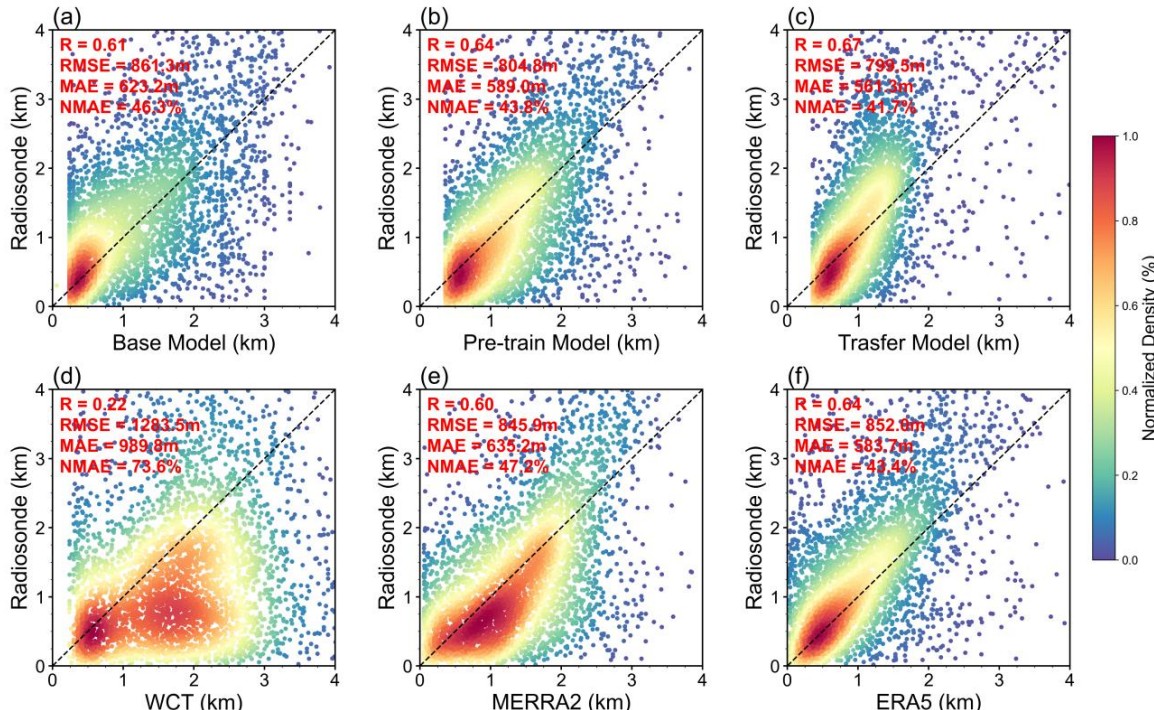

**Fig. 6.** Scatter plots comparing PBLH estimations from (a) base model, (b) pre-train model, (c) transfer trained model, (d) WCT, (e) MERRA2, and (f) ERA5 against radiosonde observations. Unlike Fig. 5, these comparisons employ direct radiosonde-derived PBLH rather than the radiosonde-constrained target labels. Statistical metrics, R, MAE, NMAE, and RMSE are present in red at the upper left of each subplot.

As results, the classical WCT shows the poorest consistence with radiosondes, with the weakest correlation (0.22) and largest MAE/NMAE (989.8 m/73.6%) among all these comparisons. Although we have previously obtained reasonable consistency between them by filtering PBLH under stable regime and separately comparing samples under cloudy and clear-sky conditions (Li et al., 2023), those statistical metrics remained inferior to the comparisons between radiosonde and MERRA2 and ERA5. However, our ResNet model significantly enhances the representation for the truth PBLH. The transfer model demonstrates marked improvements in predictive capability, exhibiting higher consistency with radiosondes than both the base and pre-train model, as well as two reanalysis datasets, with the strongest correlation (0.67) and the lowest MAE/NMAE (561.3 m/41.7%). In addition, Table 1 gives the comparisons between the WCT, base, pre-train, and transfer models and radiosondes at each daily hour. The transfer model is also superior than others at hourly scale, indicating its ability to capture more accurate diurnal variations of PBLH. transfer

**Table 1**. Statistics metrics for comparing PBLH from WCT, base model, pre-train model, and transfer model to radiosonde derived PBLH at each daytime hours, values following "±" represent the 95% confidence level.

| Local Time | | 8 | 9 | 10 | 11 | 12 | 13 | 14 | 15 | 16 | 17 | 18 |
|---|---|---|---|---|---|---|---|---|---|---|---|---|
| WCT | R | 0.17±0.08 | 0.03±0.09 | 0.22±0.12 | 0.12±0.14 | 0.28±0.11 | 0.26±0.10 | 0.21±0.09 | 0.28±0.10 | 0.14±0.10 | 0.24±0.09 | 0.18±0.10 |
| | RMSE | 1.22±0.07 | 1.45±0.08 | 1.3±0.10 | 1.16±0.10 | 1.1±0.09 | 1.25±0.08 | 1.29±0.08 | 1.28±0.08 | 1.4±0.10 | 1.25±0.08 | 1.28±0.08 |
| | MAE | 0.95±0.06 | 1.16±0.08 | 1.05±0.10 | 0.88±0.10 | 0.84±0.08 | 0.94±0.08 | 0.97±0.07 | 1.0±0.08 | 1.08±0.09 | 0.94±0.08 | 0.98±0.08 |
| | NMAE | 138.3±9.3 | 175.6±12.0 | 138.2±12.2 | 80.4±9.6 | 59.4±5.8 | 59.1±5.0 | 56.7±4.5 | 53.7±4.5 | 57.0±4.9 | 54.7±4.7 | 66.0±5.4 |
| Base | R | 0.37±0.07 | 0.37±0.08 | 0.35±0.11 | 0.62±0.09 | 0.44±0.11 | 0.55±0.08 | 0.43±0.08 | 0.52±0.08 | 0.49±0.08 | 0.5±0.08 | 0.56±0.07 |
| | RMSE | 5.58±0.31 | 5.13±0.31 | 5.41±0.42 | 6.63±0.59 | 8.09±0.62 | 9.1±0.59 | 10.21±0.62 | 10.64±0.72 | 10.95±0.74 | 10.57±0.69 | 9.09±0.59 |
| | MAE | 0.38±0.04 | 0.35±0.03 | 0.41±0.04 | 0.52±0.06 | 0.62±0.06 | 0.68±0.06 | 0.79±0.06 | 0.83±0.07 | 0.85±0.07 | 0.8±0.06 | 0.67±0.06 |
| | NMAE | 54.3±5.0 | 52.9±5.1 | 53.9±5.6 | 47.2±5.2 | 43.9±4.3 | 42.6±3.7 | 45.9±3.5 | 44.4±3.8 | 44.7±3.8 | 46.8±3.9 | 44.7±4.1 |
| Pre-Model | R | 0.25±0.08 | 0.35±0.08 | 0.45±0.10 | 0.49±0.12 | 0.6±0.08 | 0.63±0.06 | 0.56±0.07 | 0.61±0.07 | 0.56±0.08 | 0.58±0.07 | 0.53±0.07 |
| | RMSE | 5.19±0.30 | 5.28±0.32 | 6.59±0.51 | 6.57±0.59 | 6.68±0.51 | 8.12±0.52 | 9.02±0.55 | 9.67±0.66 | 10.25±0.70 | 9.54±0.62 | 9.48±0.61 |
| | MAE | 0.39±0.02 | 0.43±0.03 | 0.54±0.05 | 0.5±0.06 | 0.5±0.05 | 0.59±0.06 | 0.65±0.05 | 0.71±0.07 | 0.77±0.07 | 0.71±0.06 | 0.71±0.06 |
| | NMAE | 57.2±4.2 | 64.7±4.2 | 70.7±6.0 | 45.8±5.4 | 35.2±3.7 | 37.0±3.5 | 38.1±3.3 | 38.2±3.6 | 40.6±3.7 | 41.6±3.6 | 47.7±4.2 |
| Tra-Model | R | 0.42±0.07 | 0.37±0.08 | 0.49±0.09 | 0.57±0.10 | 0.52±0.09 | 0.66±0.06 | 0.6±0.06 | 0.62±0.07 | 0.64±0.06 | 0.57±0.07 | 0.53±0.07 |
| | RMSE | 4.68±0.27 | 4.7±0.28 | 4.26±0.34 | 6.18±0.56 | 7.53±0.58 | 8.15±0.52 | 9.18±0.56 | 10.17±0.69 | 10.1±0.69 | 10.08±0.65 | 9.21±0.60 |
| | MAE | 0.35±0.03 | 0.36±0.03 | 0.33±0.03 | 0.44±0.06 | 0.53±0.06 | 0.56±0.06 | 0.65±0.06 | 0.76±0.07 | 0.76±0.07 | 0.75±0.07 | 0.67±0.06 |
| | NMAE | 50.4±3.9 | 54.1±4.2 | 43.9±4.2 | 40.5±5.5 | 37.4±4.5 | 35.3±3.7 | 38.0±3.5 | 41.0±3.8 | 39.9±3.6 | 43.5±3.8 | 45.3±4.2 |

Since the pre-train model using pseudo-labels constrained by MERRA2 PBLH, the statistical metrics may inject reanalysis systematic deviation into the learned representation. Residual deviation correlation analysis was carried out to quantify this impacts. We calculated the residual biases between radiosonde-based PBLH and others (MERRA2, base model, pre-train model, and transfer model estimated PBLH), and compared them against the residual bias of MERRA2 (Fig. 7). The R and $R^2$ in Fig. 7a indicates that the base model itself incorporates some bias information from MERRA2 (15%), this may be due to the meteorological features inputted the model are generated from MERRA2. However, the system's deviation of MERRA2 is deeply embedded in the pre-train model, 80% of its residual can be explained by the MERRA2 deviation. Despite fine-tuning the model's weights using radiosonde labels can mitigate this impacts, with $R^2$ dropped to 0.38, it is undeniable that the transfer model still introduces a certain deviation information from MERRA2. Herein, we quantified the reduction rate of transfer training by a function: $\left(R_{pre}^2 - R_{tra}^2\right) / \left(R_{pre}^2 - R_{base}^2\right) \times 100\%$, and found the reduction rate reaches 64.6%, suggesting that fine-tuning can effectively weaken the generalization impact of the MERRA2 deviation.

Compared to the base model, our transfer model has better overall performance (Fig. 6), suggesting that the model has achieved bias mitigation while retaining the advantages of the pre-train model. Even so, it is declare to integrate multi-source observations to reduce this impacts in future work.

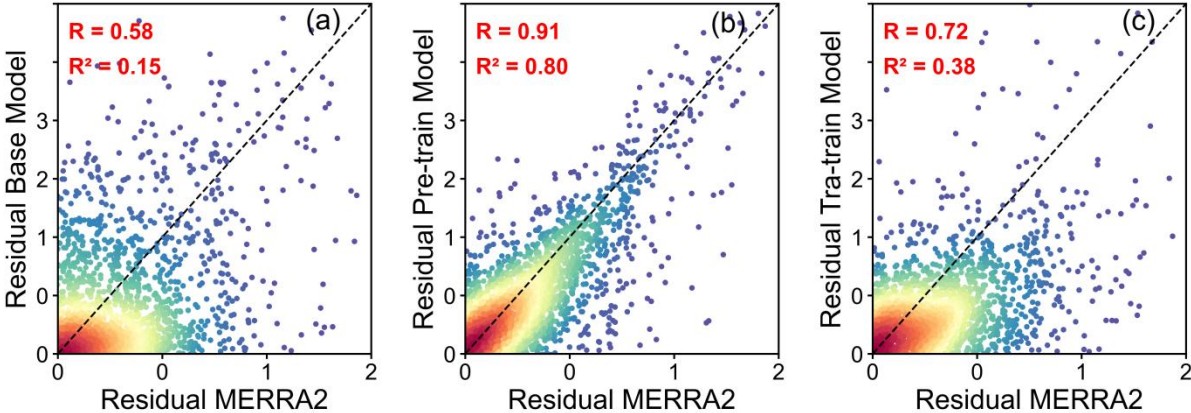

**Fig.7**. Comparisons of residual deviations from base, pre-train, transfer model against those from MERRA2 PBLH. (a) base v.s. MERRA2, (b) Pre-train v.s. MERRA2, and (c) Transfer v.s. MERRA2.

Although radiosondes are considered as ground truth and often serve as benchmarks for evaluating models and reanalysis outputs, complete quantitative consistency cannot be pursued due to mis-matches in both space and time with other datasets and discrepancies in retrieval algorithms. Instead, we can only ensure certain consistency in their spatio-temporal characteristics. The prominent consistency between the transfer model and radiosonde demonstrates the superiority of deep learning approaches and validates the rationality of our experimental design. However, it should be noted that the matchup between orbital CATS data and radiosonde sites remain relatively crude, exhibiting considerable mismatches in temporal, horizontal distance, and altitudes. Moreover, different vertical resolution of radiosondes induce uncertainties in deriving PBLH. Therefore, the PBLH differences between the transfer model compared to the sounding derived PBLH were examined in Fig. S6, with respect to their sensitivity to the matching differences in distance, time, and altitude, as well as vertical resolution of radiosonde. It can be observed that although the PBLH deviations exhibit slight dependence on time difference, distance difference, and vertical resolution,implying that the matching criteria between the radiosonde sites and CATS orbits cannot cause substantial uncertainties in this study. However, significant PBLH differences

emerge as the altitude difference increasing. This is related to the poor model performance over rugged terrain, and it also highlights the heterogeneity of PBLH over complex terrains.

4.3  transfer*Diurnal variations in near-global PBLH*

Benefiting from the unique operational characteristics of the CATS, the near-global diurnal variations in PBLH can be obtained after approximately 16 days of operation. However, due
to interference from multi-layer structures and noises in backscatter signals, diurnal variations derived by classical WCT algorithm often present non-physical fluctuations (Li et al., 2023). This study aims to extract more physically reasonable diurnal PBLH variations from CATS data using a deep learning approach. Based on theory by Stull (1988), we assumed that daytime PBLH evolution undergoes four distinct phases: morning transition (08:00–09:00),
rapid growth (10:00–14:00), maintenance (14:00–16:00), and decay in the late afternoon (17:00–18:00). Fig. 8 presents spatial distributions of PBLH for the four evolution periods derived from WCT, pre-train model, transfer model, MERRA2, and ERA5. Furthermore, Fig. S7 provides their details by highlighting the specific PBLH at each daytime hour. These results demonstrate reasonable diurnal PBLH behaviors, and they show evident differences
among different datasets or methodologies.

Similar as previous results by Li et al. (2023), the diurnal variation amplitudes derived by the WCT algorithm are severely weaken, showing no significant difference between the morning transition period and the afternoon maintenance period. In contrast, our ResNet model capture clearer diurnal patterns: lower PBLH is observed in the morning transition
period, gradually increases at the growth period, reaches its daily maximum in the maintenance stage, and then began to decline during the decay period. The transfer model exhibits some anomalous performance, such as its higher PBLH over high-altitudes and deserts during the maintenance and decay stages. We suspect that the transfer model may deviate from actual situations over these areas, as the assessments in Fig. 5 has proved the
model's ability over these areas are relatively weaker than others. Fig. S8 also shows the model's prediction biases for hourly PBLH are larger over plateaus and deserts, especially in maintenance and decay stages. On average, the MAE in high-elevation and desert is 260.2 m and 187.6 m higher than that in low-elevation and non-deserts. This partly stems from the

inherent limitation in feature extraction capability of the pre-train model over high-altitude
regions (Fig. 3). Furthermore, the scarcity of available training samples in high-altitude
regions for the transfer model can also cause substantial uncertainties in its performances.
Additionally, the transfer model predicted PBLH in the later afternoon does not significant
decay and remained notably higher than those from other methods or datasets. Fig. S9
illustrates the diurnal variations of PBLH derived from the transfer model at four seasons.
There are almost no discernible decays in PBLH during summer (JJA in the Northern
Hemisphere and SON in the Southern Hemisphere); instead, it even maintains an increasing
trend. In contrast, slight PBLH decays were observed in other seasons.

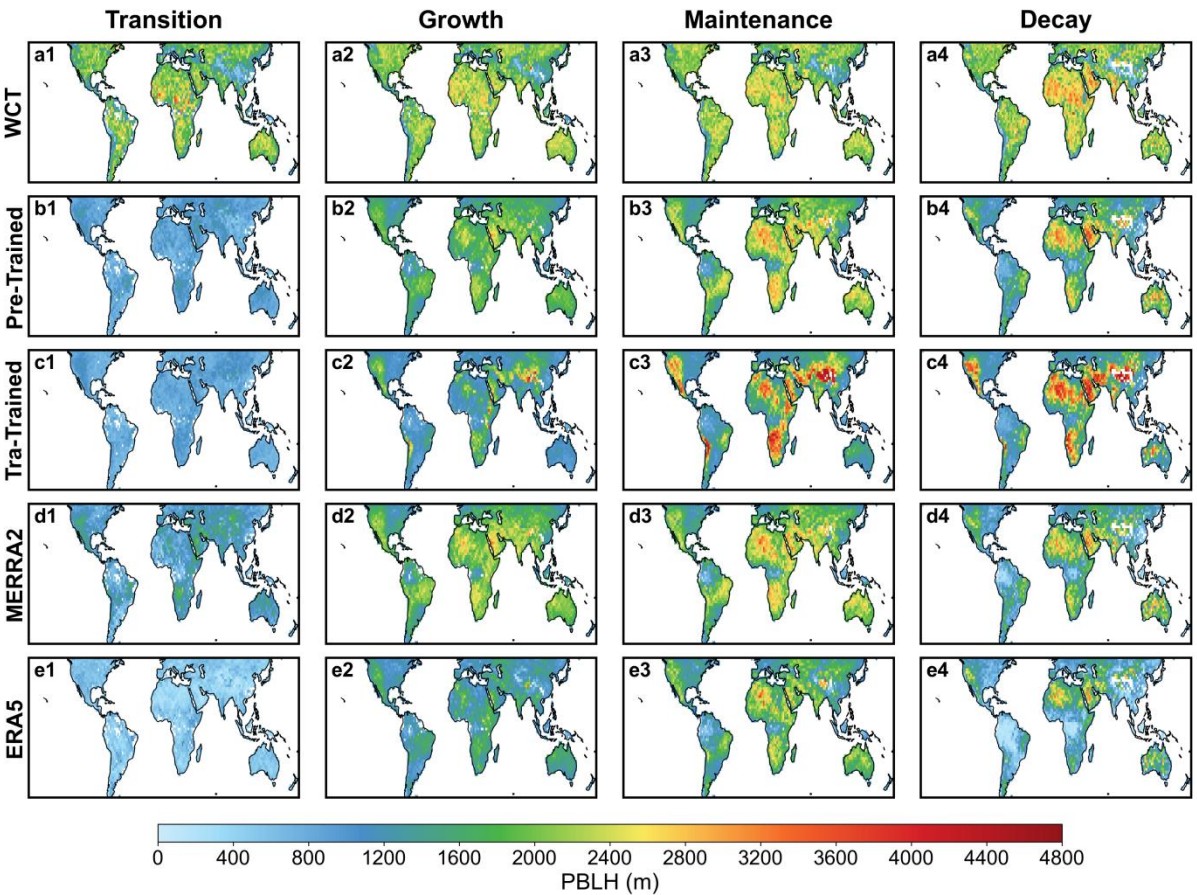

**Fig. 8.** Spatial distributions of PBLH derived from (a1-a4) WCT, (b1-b4) pre-train model, (c1-c4) transfer
model, (d1-d4) MERRA2, and (e1-e4) ERA5 during four diurnal evolution phases: morning transition,
rapid growth, maintenance, and afternoon decay.

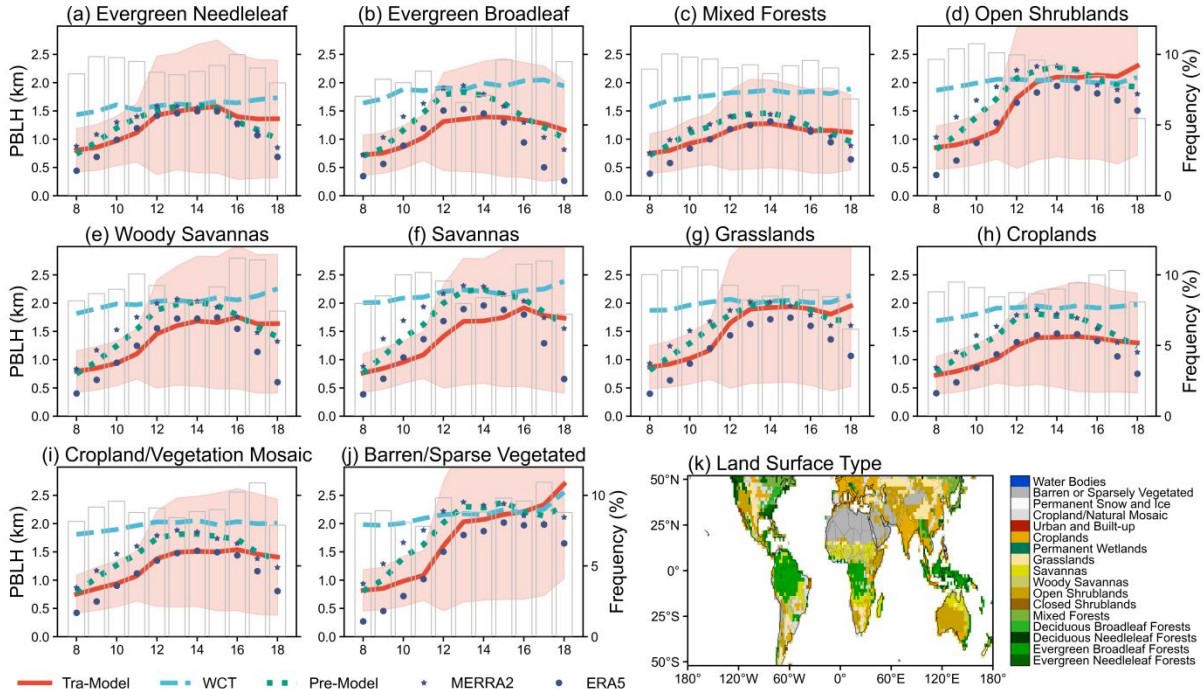

**Fig. 9**. Hourly PBLH from WCT, pre-train model, transfer model, MERRA2, and ERA5 over the major ten land cover types. The bar plots denote sampling frequency for a specified land cover at each daytime hours. (k) reveals the land cover distributions across 2° × 2° grids.

Evolution of PBLH is mainly governed by surface conditions and is highly dependent on land surface types (Li et al., 2021). To better illustrate its diurnal variation, Fig. 9 presents the hourly PBLH across ten major land surface types (derived from the three approaches and two reanalyses). The transfer model demonstrate significant improvements in capturing diurnal variations compared to WCT at most land covers, exhibiting more reasonable diurnal patterns in terms of amplitude, growth duration, and peak timing. Particularly, the model presents clearer morning growth phase and more accurate peak timing. Additionally, the model predicted PBLH exhibits a more pronounced dependence on land cover, with higher PBLH and greater diurnal amplitude observed over bare soil and shrublands compared to forests, croplands, and grassland areas. These findings are consistent with our previous observation based report (Li et al., 2021), whereas the WCT predicted PBLH exhibits much smaller deviations across different land surface types. In addition, the diurnal PBLH variation patterns (amplitude, peak timing) derived from our models aligned closely with those from the two reanalyses. Specifically, the pre-train model displays nearly identical diurnal patterns to MERRA2, while the transfer model performs more closely with ERA5 during the growth

and maintenance period. However, the transfer model predicted much higher PBLH than ERA5 during transition and decay phases.

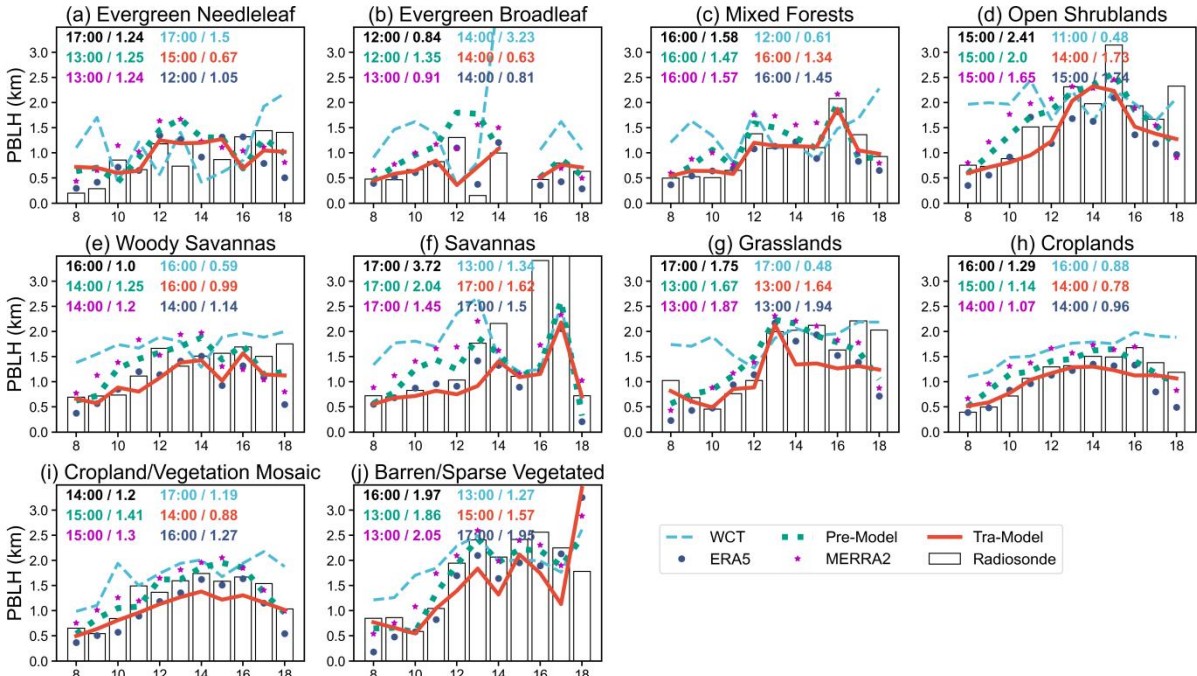

**Fig. 10**. Comparing the diurnal variations of PBLH estimated from WCT, base model, pre-train model, transfer model, MERRA2, and ERA5 to that derived from radiosondes (bar plots). text in subplots represent the peak timing and amplitude for each diurnal curve, which is formatted with "peak timing / amplitude".

As mentioned above, our transfer model derived PBLH decay in the later afternoon is not pronounced in most land covers, with PBLH magnitudes during this period being notably higher than those from the reanalyses and pre-train model. This is primarily due to reanalysis outputted PBLH is highly dependent on thermodynamic conditions and begins to decay after surface thermal flux reaches its afternoon maximum. In contrast, the model predicted PBLH is associated with backscatter of aerosols, which does not diminish synchronously with thermodynamic weakening (Wang et al., 2012). Likewise, Pearson et al. (2010) synthesized numerous studies regarding the diurnal variations of PBLH, obtaining diurnal variation curves that resembled the results from our transfer model, which suggested the credibility of the diurnal patterns predicted by our transfer model. To further support this conclusion, we qualitatively compared the diurnal PBLH patterns from WCT, three model predictions, and reanalysis outputs, to the radiosonde observations across the ten major land types (Fig. 10). Although these diurnal variations were composited from radiosonde sites at different

longitudes (potentially inducing perturbations in diurnal curves), their one-to-one matchup with other PBLH can still provide certain effective evidences. The results demonstrate that over most land covers, PBLH from the two reanalyses show more pronounced decays and lower magnitudes in the later afternoon than the radiosonde derived PBLH. Additionally, ERA5 exhibits lower PBLH than radiosonde observations during the morning transition and afternoon decay periods. Based on their diurnal variations on the seasonal scale (Fig. S10), we evaluated several metrics of their diurnal variations with the radiosonde derived PBLH, including R, MAE, and IA (index of agreement, Li et al., 2023). These metrics were calculated separately for the periods before and after the peak of the diurnal variations of sounding PBLH (Fig. S11). The results show that for those land cover types with sufficient sample sizes, the performance of the transfer model is superior to that of the base and the pre-train model, even more often superior to reanalyses. These findings indicate that our transfer model captures more reasonable diurnal patterns, and the ResNet based transfer learning approach can effectively estimate near-global PBLH from CATS data.

## 5. Conclusions

This study developed a spatially and temporally applicative ResNet learning framework to estimate near-global diurnal variations in PBLH from approximately three years of CATS lidar profiles. The proposed model demonstrates significant enhancement in estimating large-scale PBLH compared to classical algorithm. The framework is designed based on the concept that the first few peaks in WCT profiles typically capture the true PBLH, and the model is inherently proposed to identify the peak with the highest probability of representing the actual PBLH. Given the radiosonde measured PBLH samples for training a robust deep learning model are insufficient, this study adopted a transfer learning strategy. We first trained a base model using pseudo-target constrained by MERRA2 PBLH and then fine-tuned the base model on a smaller sampling dataset to generate the optimal model. This transfer model retained the strong feature extraction capabilities of the pre-train model and demonstrated considerable improvement in performance when evaluated on unseen data.

The input features for the model include remotely sensed and meteorological profiles,

geographic and temporal information, as well as surface/sky conditions. Among these, candidate PBLH derived from CATS backscatters and temperature profiles are the two dominant factors influencing model performance, collectively accounting for more than 45% of the importance scores. Their importance exhibits a distinct diurnal variation with alternating dominance: candidate PBLH primarily influences morning periods while

temperature dominates the afternoon. This alternating dominance pattern further explains the diurnal variation in model performance, with higher accuracy and lower MAE/NMAE observed during morning hours and the opposite tendencies occurred in the late afternoon. Despite these temporal fluctuations, the transfer model demonstrates overall superior performance metrics when compared against radiosondes, outperforming the results obtained

from WCT, pre-train model, MERRA2, and ERA5.

Regarding diurnal variation, the transfer model predicted PBLH exhibits clear diurnal patterns, demonstrating more reasonable diurnal amplitude, growth duration, and peak timing compared to the classical WCT algorithm. Although the model struggles to capture PBLH over high-altitude regions like the Tibetan Plateau due to insufficient training samples and

low data quality, its performances in other regions are significantly better. Particularly, the model derived diurnal PBLH variations are sensitive to land covers. PBLH over bare and shrub lands exhibit higher magnitude and larger diurnal amplitudes than that over forests, croplands, and other vegetated areas. Furthermore, the model maintains high PBLH magnitudes in the late afternoon and shows only slight decay, differing from the pronounced

decay phases of the two reanalyses derived PBLH. Even so, this non-prominent afternoon decay aligns well with radiosonde measurements, indicating its superior capability in capturing diurnal PBLH.

This study involves an initial attempt of using a deep neural network to address complex signal structure in CATS backscatter, and then to re-fine its measurement for PBLH on a

near-global scale. Although utilizing attention augmented ResNet and transfer learning strategy can effectively improve the model's capability, its performances in high-altitude regions and deserts in the morning and later afternoon periods remain poor. Future efforts would be prospected to refine the model's applicability in rugged topography or on specified land covers, integrating multi-source observations with fine-resolution meteorological data

and accurate target label are crucial for improving the model performances.

**Author contributions**

YL: Conceptualization, Methodology, Formal analysis, Investigation, Writing-original draft, Software, Validation. ZL: Software, Validation, Formal analysis, Writing-review and editing. JH: Conceptualization, Formal analysis, Methodology, Supervision, Writing-review and editing.

**Code and data availability**

Data and software used in this study are available as follows. Relevant datasets and scripts necessary to understand, evaluate, and extend the research findings reported in this paper were archived in the Zenodo repository, accessible under the DOI: 10.5281/zenodo.16907935 (https://zenodo.org/records/16907935). The IGRA V2 radiosonde data (Durre and Yin, 2008)

is available at https://www.ncei.noaa.gov/products/weather-balloon. CATS (Yourks, et al., 2016) data can be acquired from https://cats.gsfc.nasa.gov/. ERA5 (Hersbach et al., 2023) data obtained from Copernicus Climate Change Service (C3S) Climate Data Store accessible at https://cds.climate.copernicus.eu/. MERRA2 data (Gelaro et al., 2017) is archived at https://disc.gsfc.nasa.gov/datasets?keywords=MERRA-2andpage=1.


**Acknowledgements**

We acknowledge the provision of radiosonde by the National Oceanic and Atmospheric Administration (NOAA), space-borne CATS lidar by the National Aeronautics and Space Administration (NASA), ERA5 and MERRA2 reanalyses by Copernicus Climate Change

Service and NASA Global Modeling and Assimilation Office.

**Financial support**

This study was supported by the National Key Research and Development Program of China (2023YFC3706304).


**Competing interests**

The contact author has declared that none of the authors has any competing interests.

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
