# Peer review of "Improved estimation of diurnal variations in near-global PBLH through a hybrid WCT and transfer learning approach"

_EGUsphere, 2025_

## Author Comment (AC1)

**Reply on RC1:**

This study reports a new hybrid approach that combines the wavelet covariance transform (WCT) with a transfer-learning deep residual network to estimate the diurnal evolution of planetary boundary layer height (PBLH) at near-global scale from the non-sun-synchronous CATS spaceborne lidar. The proposed transfer-learning strategy is both novel and practical, effectively leveraging the large-sample coverage of reanalysis products and the high accuracy of radiosonde measurements. The methodology is sound, the experiments are comprehensive, and the results clearly demonstrate substantial performance gains over conventional algorithms. The findings are valuable for improving boundary-layer parameterizations and advancing our understanding of global PBL diurnal variability, and they fall well within AMT's scope.

However, several aspects require deeper discussion and additional evidence to further strengthen the reliability of some results and enhance the paper's scientific contribution and technical impact. I therefore recommend acceptance after minor revision.

**Specific comments and suggestions**

1. **Pseudo-label bias from MERRA-2.**
Pretraining with MERRA-2-constrained pseudo-labels inherently injects reanalysis systematic biases into the learned representation. Even after fine-tuning with 4,662 radiosonde-matched samples, residual biases may persist (as also suggested by the closer agreement of the pretrained model with MERRA-2). Please discuss and, if possible, quantify this effect and its impact on the final estimates.

**Response**: We are grateful to the reviewer for providing this profound feedback, we fully agree with the reviewer's viewpoint. We must admit that due to the limit number of samples where the radiosondes and CATS over-lapping orbits coincide, they are far from sufficient for model training. Therefore, we adopted the MERRA2 reanalysis data to pre-train a feature extractor. However, this inevitably introduced errors from the reanalysis data. In fact, during the parameter fine-tuning process of the transfer training, we did not merely adjust the weights of the fully connected layers, but released the weights of the residual structure of the third layer, allowing the model to learn as many features as possible from the real labels. Since the pre-trained model using pseudo-labels constrained by MERRA2 PBLH, the statistical metrics may inject reanalysis systematic deviation into the learned representation. Residual deviation correlation analysis was carried out to quantify this impacts. We calculated the residual deviations between radiosonde-based PBLH and others (MERRA2, base model, pre-train model, and transfer model estimated PBLH), and compared them against the residual deviations of MERRA2 (Fig. 7). The R and $R^2$ in Fig. 7a indicates that the base model itself incorporates some bias information from MERRA2 (15%), this may be due to the meteorological features inputted the model are generated from MERRA2. However, the system's deviation of MERRA2 is deeply embedded in the pre-trained model, 80% of its residual can be explained by the MERRA2 deviation. Despite fine-tuning the model's weights using radiosonde labels can mitigate this impacts, with R2 dropped to 0.38, it is undeniable that the transfer model still introduces a certain deviation information from MERRA2. Herein, we quantified the reduction rate of transfer training by a function: , and found the reduction rate reaches 64.6%, indicating that fine-tuning can effectively weaken the generalization impact of the MERRA2 deviation. Compared to the base model, our transfer model has better overall performance (Fig. 6), suggesting that the model has achieved bias mitigation while retaining the advantages of the pre-trained model. Even so, it is declare to integrate multi-source observations to reduce this impacts in future work.

[Figure]

Fig.7. Comparisons of residual deviations from base, pre-train, transfer model against those from MERRA2 PBLH. (a) base v.s. MERRA2, (b) Pre-train v.s. MERRA2, and (c) Transfer v.s. MERRA2.

2. **Train/validation/test independence.**

The manuscript states that 2016 data are used for pretraining and that the transfer stage uses a 4,000/662 split, but it does not clarify whether the test set is strictly separated by station and time window. To avoid information leakage from adjacent or same-station samples, please clarify the split strategy and consider a station- and season-stratified (or leave-one-site-out) evaluation.

**Response**: We are grateful to the reviewer for pointing out this issue, which is very helpful for enhancing the credibility of our work. In our sample selection process, we indeed overlooked this aspect and only selected 662 samples through random sampling, which obviously posed a risk of data leakage. Therefore, in the revised manuscript, we re-screened the testing samples. Due to the strict temporal and spatial isolation when matching the CATS profiles of different orbits with the sounding stations, we believe that the risk of data leakage is extremely low. In the revised version, we only processed the samples from the same orbit that were relatively close to each other. Specifically, if the distance between the sounding sites matched to the same CATS orbit is less than 300 km, we consider there is a risk of data leakage in the spatial dimension. This type of sample will not be included in the testing set but will all be placed into the training set. Additionally, in the revised manuscript, the radiosonde sites in the ocean-island were removed. To ensure that there were still sufficient samples available for training the transfer model after eliminating the ocean-based radiosondes, we increased the matching distance between CATS orbits and the sounding sites from less than 150 km to less than 200 km. Eventually, we obtained 5008 matching samples, of which 750 (15%) were used for the testing. Based on this new test set, we retrained the transfer model. The results showed that the accuracies of the base model, the pre-train model, and the new transfer model on the new test set were 60.0%, 59.2%, and 68.3% respectively. This indicates that adopting a transfer learning strategy in this work is still appropriate. The new transfer model we trained did not change the key findings, but some of the results are different from those in our original manuscript. Therefore, we updated most figures in revised manuscript (from Fig.3 to Fig.10) when adopting newly trained transfer model and modified some of the conclusions.

3. **Statistical vs. physical consistency of the afternoon decay.**

You conclude that the model exhibits a weaker afternoon decay and better agreement with radiosondes, while morning correlations are slightly lower but accuracy is higher (smaller bias). I recommend a joint assessment of **statistical consistency** (e.g., R, MAE) and **physical consistency** (e.g., decay rate after the

diurnal peak and the timing of the peak). This would help reconcile performance metrics with expected PBL diurnal physics.

**Response**: The reviewer's comment is very insightful, apart from evaluating the statistical consistency of the model's prediction results, we also calculated the physical consistency. That is, from the perspective of diurnal variations, we calculated several metrics such as R, MAE and IA (index of agreement) during two phases (growth and decay). However, since the diurnal PBLH variations on the ten land cover in Fig. 10 are composited from PBLH of different longitudes. These only 11 PBLH values are insufficient to calculate statistically significant indicators. Therefore, in the revised version, we also calculated the seasonally diurnal variations over these land cover types (Fig.S10), such that the amount of PBLH data is sufficient for us to calculate these statistical indicators. In the revised manuscript, we obtained the peak timing of the diurnal variation of the PBLH in the sounding, and we took this time as the benchmark. The period before the peak was considered the growth phase, and the period after the peak was regarded as the decay phase. Then, we separately calculated the R, MAE and IA indicators for different stages (Fig.S11). The results show that for those land cover types with sufficient sample sizes, the performance of the transfer model is superior to that of the base and the pre-train model, even more often superior to reanalyses.This will further enhance our understanding of the model's ability to predict the diurnal PBLH variations from CATS.

[Figure]

Fig. S10. Seasonally diurnal variations of PBLH on ten major land cover types, which are derived from radiosondes (columns), pre-train model (cyan dashed), transfer model (red solid), ERA5 (solid circle), and MERRA2 ('+').

[Figure]

Fig. S11. R, MAE, and IA (index of agreement) between model/reanalysis predicted and radiosonde derived PBLH on ten major land cover types.

4. **Quantifying uncertainty over complex terrain and deserts.**

The diagnosis for reduced performance over high-elevation and desert regions is reasonable, but quantitative uncertainty information is missing. Please provide uncertainty maps and/or tables, for example seasonal and hourly MAE/bias boxplots specifically for these regions.

**Response**: Yes, we agree with the concerns raised by the reviewer. The previous manuscript only qualitatively pointed out that the model performance was poor in complex terrains and desert areas, but did not quantify the uncertainty of the model. To present the uncertainty of the model in these areas more intuitively, we followed the reviewers' suggestions and provided the comparisons for MAE and NAME (normalized mean absolute error) over these areas. The results show that the PBLH is much higher in high-elevation and desert regions than others (Fig. S8), especially in the maintenance and decay stages. On average, the MAE in high-elevation and desert is 260.2 m and 187.6 m higher than that in low-elevation and non-deserts.

[Figure]

Fig. S8. Comparisons for MAE and NMAE of PBLH between high-altitude and low-altitude (a), and between deserts and non-deserts (b) at each daytime hour.

5. **Implications from feature importance.**

Since "candidate PBLH" and temperature together contribute >50% of the importance, with LST/elevation next and TAB/WCT shape metrics relatively low, the conclusions should more explicitly articulate the implication for classical algorithms: rather than further refining profile-shape heuristics, incorporating thermodynamic and terrain-related diagnostics appears more beneficial.

**Response**: Thanks to the reviewers' comments, machine learning methods have the ability to pick out helpful features from complex signals. However, our results show that if the model integrates TAB and

WCT profiles, it cannot learn any useful information from them. This also means that improving the classical algorithm points out some directions. If only the signal and structure of the signal profile are considered, the retrieval results will not be improved. Instead, corresponding meteorological conditions such as thermodynamic, moisture, and dynamic factors should be more incorporated. We have added some further discussions in Section 3.2.

6. **Figure and table clarity.**

In Figure 10, please annotate each land-cover curve with the peak time and amplitude to aid interpretation. For Table 1 (hourly R/MAE/RMSE), consider adding 95% confidence intervals or bootstrap-based uncertainty bands.

**Response**: Following the reviewer's suggestion, we have added peak time and amplitude indicators in Figure 10 and supplemented the confidence intervals in Table 1. We believe these modifications will make the manuscript more intuitive and readable.

[Figure]

Fig. 10. Comparing the diurnal variations of PBLH estimated from WCT, base model, pre-train model, transfer model, MERRA2, and ERA5 to that derived from radiosondes (bar plots). text in subplots represent the peak timing and amplitude for each diurnal curve, with a format of "peak timing / amplitude".

Table 1. Statistics metrics for comparing PBLH from WCT, base model, pre-train model, and transfer model to radiosonde derived PBLH at each daily hours, values following "±" represent the 95% confidence level.

| Local Time | | 8 | 9 | 10 | 11 | 12 | 13 | 14 | 15 | 16 | 17 | 18 |
|---|---|---|---|---|---|---|---|---|---|---|---|---|
| WCT | R | 0.17±0.08 | 0.03±0.09 | 0.22±0.12 | 0.12±0.14 | 0.28±0.11 | 0.26±0.10 | 0.21±0.09 | 0.28±0.10 | 0.14±0.10 | 0.24±0.09 | 0.18±0.10 |
| | RMSE | 1.22±0.07 | 1.45±0.08 | 1.3±0.10 | 1.16±0.10 | 1.1±0.09 | 1.25±0.08 | 1.29±0.08 | 1.28±0.08 | 1.4±0.10 | 1.25±0.08 | 1.28±0.08 |
| | MAE | 0.95±0.06 | 1.16±0.08 | 1.05±0.10 | 0.88±0.10 | 0.84±0.08 | 0.94±0.08 | 0.97±0.07 | 1.0±0.08 | 1.08±0.09 | 0.94±0.08 | 0.98±0.08 |
| | NMAE | 138.3±9.3 | 175.6±12.0 | 138.2±12.2 | 80.4±9.6 | 59.4±5.8 | 59.1±5.0 | 56.7±4.5 | 53.7±4.5 | 57.0±4.9 | 54.7±4.7 | 66.0±5.4 |
| Base | R | 0.37±0.07 | 0.37±0.08 | 0.35±0.11 | 0.62±0.09 | 0.44±0.11 | 0.55±0.08 | 0.43±0.08 | 0.52±0.08 | 0.49±0.08 | 0.5±0.08 | 0.56±0.07 |

| | | | | | | | | | | | |
|---|---|---|---|---|---|---|---|---|---|---|---|
| | RMSE | 5.58±0.31 | 5.13±0.31 | 5.41±0.42 | 6.63±0.59 | 8.09±0.62 | 9.1±0.59 | 10.21±0.62 | 10.64±0.72 | 10.95±0.74 | 10.57±0.69 | 9.09±0.59 |
| | MAE | 0.38±0.04 | 0.35±0.03 | 0.41±0.04 | 0.52±0.06 | 0.62±0.06 | 0.68±0.06 | 0.79±0.06 | 0.83±0.07 | 0.85±0.07 | 0.8±0.06 | 0.67±0.06 |
| | NMAE | 54.3±5.0 | 52.9±5.1 | 53.9±5.6 | 47.2±5.2 | 43.9±4.3 | 42.6±3.7 | 45.9±3.5 | 44.4±3.8 | 44.7±3.8 | 46.8±3.9 | 44.7±4.1 |
| **Pre-Model** | R | 0.25±0.08 | 0.35±0.08 | 0.45±0.10 | 0.49±0.12 | 0.6±0.08 | 0.63±0.06 | 0.56±0.07 | 0.61±0.07 | 0.56±0.08 | 0.58±0.07 | 0.53±0.07 |
| | RMSE | 5.19±0.30 | 5.28±0.32 | 6.59±0.51 | 6.57±0.59 | 6.68±0.51 | 8.12±0.52 | 9.02±0.55 | 9.67±0.66 | 10.25±0.70 | 9.54±0.62 | 9.48±0.61 |
| | MAE | 0.39±0.02 | 0.43±0.03 | 0.54±0.05 | 0.5±0.06 | 0.5±0.05 | 0.59±0.06 | 0.65±0.05 | 0.71±0.07 | 0.77±0.07 | 0.71±0.06 | 0.71±0.06 |
| | NMAE | 57.2±4.2 | 64.7±4.2 | 70.7±6.0 | 45.8±5.4 | 35.2±3.7 | 37.0±3.5 | 38.1±3.3 | 38.2±3.6 | 40.6±3.7 | 41.6±3.6 | 47.7±4.2 |
| **Tra-Model** | R | 0.42±0.07 | 0.37±0.08 | 0.49±0.09 | 0.57±0.10 | 0.52±0.09 | 0.66±0.06 | 0.6±0.06 | 0.62±0.07 | 0.64±0.06 | 0.57±0.07 | 0.53±0.07 |
| | RMSE | 4.68±0.27 | 4.7±0.28 | 4.26±0.34 | 6.18±0.56 | 7.53±0.58 | 8.15±0.52 | 9.18±0.56 | 10.17±0.69 | 10.1±0.69 | 10.08±0.65 | 9.21±0.60 |
| | MAE | 0.35±0.03 | 0.36±0.03 | 0.33±0.03 | 0.44±0.06 | 0.53±0.06 | 0.56±0.06 | 0.65±0.06 | 0.76±0.07 | 0.76±0.07 | 0.75±0.07 | 0.67±0.06 |
| | NMAE | 50.4±3.9 | 54.1±4.2 | 43.9±4.2 | 40.5±5.5 | 37.4±4.5 | 35.3±3.7 | 38.0±3.5 | 41.0±3.8 | 39.9±3.6 | 43.5±3.8 | 45.3±4.2 |

**7. WCT dilation sensitivity.**

Although 480 m is identified as the optimal dilation, it would be helpful to include in the Supplement a systematic comparison table showing (i) five-peak hit rates under different dilation factors and (ii) "largest-peak only" vs. "multi-peak candidate" performance.

**Response**: We agree with the reviewer's opinion. Although Figure 1 has confirmed that a 480 m dilation factor is reasonable for the CATS signal, it is still unknown whether there are differences in the hit rates of multiple WCT peaks under different dilation factors. We followed the reviewer's suggestion and tested the hit rates of five peaks for PBLH under different dilation factors in the supplementary materials (Table S1), and compared the differences between the multiple peaks hit rate and the hit rate of largest-peak. It is worth noting that in the calculation of the hit rate for multiple peaks, there are some differences in the accuracy of the first peak compared to those shown in Fig. 1a in the manuscript. This is because our multi-peak hit rate is looking for the peak that is closest to the true value. The largest-peak hit rate only considers an accurate if the distance between the position of the maximum peak and the true PBLH value is less than the set threshold. For the multi-peak hit rate, the total hit rate is the highest when the dilation factor is 480 m. Therefore, it can be concluded that selecting 480 m as the dilation factor in this work is optimal.

**Table S1**. Accuracy (%) and mean absolute error (m) for the first five peaks with different dilation factors range from 240 m to 960 m.

| | Dialation (m) | 240 | 360 | 480 | 600 | 720 | 840 | 960 |
|---|---|---|---|---|---|---|---|---|
| **Accuracy (%)** | First Peak | 29.67 | 32.57 | 35.77 | 37.8 | 38.77 | 39.18 | 38.41 |
| | Second Peak | 17.58 | 17.21 | 17.07 | 16.16 | 16.92 | 16.72 | 15.65 |
| | Third Peak | 13.87 | 14.23 | 13.62 | 13.31 | 13.01 | 12.25 | 10.11 |
| | Fourth Peak | 14.68 | 12.91 | 11.16 | 12.04 | 9.1 | 4.27 | 0.56 |
| | Fifth Peak | 12.65 | 11.89 | 12.55 | 6.55 | 1.37 | 0 | 0 |
| | Total | 88.45 | 88.81 | 90.17 | 85.86 | 79.17 | 72.42 | 64.73 |
| **MAB (m)** | First Peak | 201.1 | 212.6 | 225.8 | 240.9 | 255.6 | 271.3 | 278.6 |
| | Second Peak | 225.2 | 245.8 | 255.0 | 271.5 | 280.8 | 298.6 | 300.2 |
| | Third Peak | 224.8 | 215.5 | 233.8 | 252.9 | 294.9 | 300.5 | 305.6 |
| | Fourth Peak | 214.9 | 225.0 | 232.0 | 248.7 | 244.3 | 250.4 | 236.8 |
| | Fifth Peak | 212.3 | 220.7 | 241.4 | 217.1 | 289.7 | - | - |

---

## Author Comment (AC2)

**Reply on RC2**:

The manuscript presents a deep learning framework using an attention-augmented ResNet with transfer learning to estimate diurnal variations of near-global planetary boundary layer height (PBLH) from CATS lidar, explicitly addressing multi-layer structures in spaceborne backscatter profiles. The topic is timely, and the approach is interesting and potentially impactful. Please see the detailed comments below.

Specific comments:

Line 129: Please spell out the date as "January 10" rather than using an abbreviation, for consistency with the rest of the manuscript.
**Response**: The abbreviation has been expanded as "January 10" in the revised manuscript.

Lines 205–210: The accuracy metric is defined as the fraction of predictions within 500 m of radiosonde PBLH. While 500 m is a reasonable tolerance for some regimes, it can be relatively large for diurnal PBLH over land. To demonstrate robustness, please justify the choice of the 500 m threshold or provide a sensitivity analysis showing how key conclusions change with the tolerance chosen.
**Response**: We are grateful to the reviewer for pointing out this deficiency. Using a fixed threshold to determine the accuracy of the model is indeed too arbitrary, especially for this PBLH with obvious diurnal variations. Therefore, we conducted some sensitivity tests on the selection of the threshold, ranging from 300 m to 700 m. As shown in Table S2, it can be observed that as the threshold increases, the hit rates of all five peaks increase linearly. The conclusion in lines 205–210 of our manuscript is mainly to reveal that using multiple WCT peaks can better capture the true PBLH, providing theoretical support for the subsequent model construction; rather than to explain the deviation between these peaks and the true PBLH. Therefore, it can be said that the selection of the threshold does not affect the main conclusion of this work. In fact, choosing 500 m as the threshold is basically consistent with the dilation factor of 480 m for calculating the WCT profiles in this work, and it is almost the uncertainty range of the WCT retrieval algorithm. In the revised manuscript, we provided a demonstration of the results of these sensitivity tests.

**Table S2**. Hit rate (%) of the first five peaks when setting different threshold for calculating accuracy.

| Threshold (m) | 300 | 400 | 500 | 600 | 700 |
|---|---|---|---|---|---|
| First Peak | 30.39 | 33.59 | 35.77 | 36.99 | 38.11 |
| Second Peak | 13.31 | 15.8 | 16.97 | 17.78 | 18.19 |
| Third Peak | 11.38 | 12.65 | 13.62 | 14.23 | 14.63 |
| Fourth Peak | 10.26 | 11.38 | 12.55 | 12.75 | 13.06 |
| Fifth Peak | 8.74 | 9.6 | 10.16 | 10.57 | 10.87 |

Line 269: Please specify the interpolation method used to map MERRA2 meteorological profiles onto the 84 CATS bins.
**Response**: We sincerely apologize for the omissions in processing the MERRA-2 reanalysis vertical profiles in the manuscript. The meteorological profiles include temperature, relative humidity, and wind speeds obtained from MERRA2 reanalysis, which were first matched with each CATS orbit, following inverse distance weight for spatial matchup and most proximity for temporal matchup. And then the

spatio-temporally matched MERRA2 profiles were vertically aligned to 84 CATS bins using a linear interpolation method.

Lines 293–296: In the transfer-learning stage, the transfer-training set comprises 4,000 samples and the remaining 662 samples serve as a common test set. Please describe how you minimized spatial and temporal leakage between training and test sets. For example, indicate whether you used station-wise or region-wise splits, any temporal separation, and provide summaries/maps of the train/test distributions to verify independence.

**Response**: We fully agree with the reviewer's concerns regarding data leakage. In the process of choosing testing samples, we indeed overlooked this aspect and only selected 662 samples through random sampling, which obviously posed a risk of data leakage. Therefore, in the revised manuscript, we re-screened the testing samples. Due to the strict temporal and spatial isolation when matching the CATS profiles of different orbits with the sounding stations, we believe that the risk of data leakage is extremely low. In the revised version, we only processed the samples from the same orbit that were relatively close to each other. Specifically, if the distance between the sounding sites matched to the same CATS orbit is less than 300 km, we consider there is a risk of data leakage in the spatial dimension. This type of sample will not be included in the testing set but will all be placed into the training set. Additionally, in the revised manuscript, the radiosonde sites in the ocean-island were removed. To ensure that there were still sufficient samples available for training the transfer model after eliminating the ocean-based radiosondes, we increased the matching distance between CATS orbits and the sounding sites from less than 150 km to less than 200 km. Eventually, we obtained 5008 matching samples, of which 750 (15%) were used for the testing. Based on this new test set, we retrained the transfer model. The results showed that the accuracies of the base model, the pre-trained model, and the new transfer model on the new test set were 60.0%, 59.2%, and 68.3% respectively. This indicates that adopting a transfer learning strategy in this work is still appropriate. The new transfer model we trained did not change the key findings, but some of the results are different from those in our original manuscript. Therefore, we updated most figures in revised manuscript (from Fig.3 to Fig.10) when adopting newly trained transfer model and modified some of the conclusions.

Line 348: You indicate ocean profiles were removed during pre-training due to limited radiosonde matchups, yet Fig. 5 shows results over oceanic areas. Please clarify whether the model was trained only on land but applied over oceans at inference.

**Response**: Our pre-train model was only trained on land and excluded the grid points in the ocean. In Fig. 5 of manuscript, there are still some results on the ocean. Actually, these ocean-based results are not from the ocean surface but are the results of matching with the sounding sites on islands. To enable our model to have more possible samples for training, we took these island radiosondes into our work. However, we must admit that our work overlooked this point raised by the reviewer. The CATS data matched to the island-based radiosonde could potentially come from the ocean. Therefore, we re-examined the samples that matched these sounding sites and found that some of the samples had a surface type of water body, which contradicted the settings of the pre-trained model. In the revised manuscript, we removed all the samples with a land type of water body, but this would further reduce the already insufficient training dataset. To ensure a sufficient sample size as much as possible, we therefore increased the spatial range of CATS matching radiosondes from 150 km to 200 km (as the model's prediction biases show less dependence on matchup distance). Ultimately, we generated 5008 matching samples, and we chosen 750 samples (~15%) from them as the common testing set for both the pre-train and transfer train model.

Lines 372–376: Please elaborate on why the model performs more poorly from April to September. If available, add supporting analyses or references.

**Response**: Thanks the reviewer for pointing this out. The poorer performance of the model in the months of April to September mainly has two reasons. As the poorer performances are primarily sourced from the Northern Hemisphere, it can be concluded that the model's representation in spring and summer seasons were somewhat weaker than that in autumn and winter. For the spring and summer seasons, the atmosphere is vigorous, accompanied by more convective activities. This leads to more complex aerosol structures (more noised CATS signal), but also limits the representation ability of MERRA2. In contrast, the atmosphere is more stagnant, and the aerosol structure is simpler (Li et al., 2025). Additionally, our assessment is mainly based on absolute deviation. The higher PBLH magnitude in the spring and summer seasons will make the assessment worse. When considering relative deviation (NMAE, Fig. 3a), the performance improves somewhat, but it is still slightly poorer than that in the autumn and winter seasons.

**References**: Li, Y., He, J., Ren, Y., and Wang, H. (2025). Aerosol-PBL relationship under diverse meteorological conditions: Insights from satellite/radiosonde measurements in North China. Atmospheric Research, 321, 108125.

Lines 380–381: The permutation importance approach is appropriate, but shuffling individual features across samples in a sequence task can yield unrealistic feature combinations when predictors are correlated (e.g., temperature and local time).

**Response**: The reviewer's comments are very insightful. We carefully considered the issue pointed out by the reviewers. It is objectively true that by randomly shuffling individual features, some false correlations between features can be generated. In our manuscript, we calculate the importance of features by shuffling individual features across samples. However, we do not randomly change the order of a single feature completely; instead, we perform block-level shuffling using 84 bins (one sample contains 84 vertical layers). Each shuffling is only performed between blocks, and the order of this feature within a block is not shuffled. That is to say, these features retain their values as the height changes. And shuffling between blocks, each sample is independent (from different times and locations), and we believe this approach can effectively alleviate the problem pointed out by the reviewer. In this work, what we have shuffled is the sequence between blocks, rather than the internal structure of the blocks. In each block of samples, the local time, longitude, latitude, and altitude have the same value for 84 bins. As the reviewer pointed out, there is indeed a correlation between temperature and local time, but when the temperature feature is shuffled at the block-level, the temperature layers of the 84 vertical layers are replaced by the values of other samples. It can be understood that the sample is from other different locations or different seasons, and the physical correlation between temperature and local time is still reasonable. Of course, using permutation importance inevitably leads to some false feature combinations. We just minimize this impact as much as possible. We hope to hear further feedback from the reviewer. If necessary, we may replace it with another more suitable feature importance estimation method.

Lines 455–457: Given that absolute PBLH magnitudes vary substantially across regions and seasons, please report relative bias metrics in addition to absolute errors. This will better reflect performance where PBLH is small or large.

**Response**: Thank to the reviewer for your suggestion. Using an absolute PBLH deviation can only indicate a certain aspect of the model's performance and cannot represent the complete performance of the actual reaction model. Following to the reviewer's suggestion, we have added curves of relative deviations in Figs. 3a-b, Fig. 3f, and Fig. 5a-d. Compared to the absolute deviation, the changes in relative deviation are more gradual, which also reflects that the model's performance has higher temporal and spatial consistency. It is smaller in both diurnal variations and spatial distributions compared to the absolute deviation. Additionally, we have calculated the relative deviation indicators spatially. Although the spatial distribution is more uniform, there are still large deviations in desert areas and high-altitude regions. This reflects that the poor performance of the model in these areas.

[Figure]

Fig. 3. Assessment of the pre-train model. (a-b) give the accuracy (column), MAE (black solid line) and NMAE (red solid line) at monthly and hourly scale, respectively; (c-f) denote the spatial distributions of accuracy, MAE, R2, and NMAE, respectively.

[Figure]

Fig. 5. Performance comparisons of the WCT, base, pre-train and transfer model against radiosonde constrained target labels. (a, c, e) show the spatial distributions of accuracy, MAE, and NMAE for transfer model,   (b, d, f) display the diurnal variations of these metrics for WCT (column), base (back dash), pre-train (blue dash), and transfer (red dash) models.

Line 584: Please specify the source of the land surface type categories used.

**Response**: We appreciate the question raised by the reviewer. In fact, the land surface type data used in this work is all derived from the CATS dataset, which integrates the surface categories from MODIS. These data are one-to-one corresponding to each CATS profile and no additional processing is required. We provide further illustration of the surface type categories in the Section 3.1. In addition, the surface classifications shown in Figs. 9 and 10 in the manuscript are also based on them.

Please review the manuscript for tense consistency.

**Response**: We have carefully reviewed the entire manuscript and revised all the inconsistent tenses to ensure tense consistency throughout the paper. We have followed academic writing conventions for tense usage: the past tense is used to describe our experimental procedures and results, the present tense is used to state general scientific facts and the purpose of this study, and the present perfect tense is used to summarize the previous research progress. All the revisions are marked in red in the revised manuscript.